# Energy-Use and Indoor Thermal Performance in Junior High School Building after Air-Conditioning Installation with the Private Finance Initiative

**Dian Sekartaji [1],\*, Yuji Ryu [1], Didit Novianto [2], Kazuma Eto [1] and Weijun Gao [1]**

[1] Department of Architecture, Faculty of Environmental Engineering, The University of Kitakyushu, Kitakyushu 808-0135, Japan
[2] Department of Architecture, Civ-Plan, Institut Teknologi Sepuluh Nopember, Surabaya 60111, Indonesia
\* Correspondence: d-sekartaji@kitakyu-u.ac.jp; Tel.: +093-695-3310

**Abstract:** Japan's government has adopted the "Private Finance Initiative (PFI)" as a project method for monitoring "air-conditioning (AC)" performance after AC installation projects to overcome heatstroke increase in schools during the summer. However, this project was conducted long after schools were built, which raises the question: what is the AC "energy-use (EU)" and how comfortable will the classroom be when it is installed without going through the planning stage? Minimizing AC EU while keeping indoor thermal comfort is the main concern for low-carbon building design technology development. This research aims to evaluate the AC EU and summer indoor thermal comfort in classrooms by position and zone. This research method analyzes PFI monitoring data, field measurement data, and questionnaires with sensitivity analysis. It found that AC EU in the summer was higher than in the winter. In addition, the AC setting temperatures in the summer (cooling) were below the government-recommended value of 28 °C. Although the indoor thermal comfort percentage in the summer had reached 75.3%, there was a seating position with a smaller comfort percentage than others. The result further shows that most students felt "neutral". However, the number of students who felt "slightly cool" and "cool" were more than those who felt "slightly warm" and "warm".

**Keywords:** energy use; indoor thermal environment; PFI; classrooms; air-conditioning; low-carbon building; junior high school

## 1. Introduction

Extreme air temperature caused by urban heat islands in the summer leads to thermal stress and causes an increase in the number of heatstroke patients in Japan [1]. In recent years, about 5000 cases of heatstroke have occurred every year in Japan's elementary schools, junior high schools, high schools, and other educational facilities, exceeding 7000 cases in 2018 [2]. For this reason, the "Ministry of Education, Sports, Science, and Technology, Japan (MEXT)" has allocated a special local grant for "air-conditioning (AC)" equipment installation in school facilities [3]. As a result, AC installation in typical classrooms is increasing rapidly in public elementary and junior high schools in Japan nationwide, from 6.2% in 2004 to 93.0% in September 2020 [4]. As we know, buildings are one of the largest energy consumers across the world [5], and "heating, ventilation, and air-conditioning (HVAC)" is the largest energy end use in buildings both in the residential and non-residential sector [6]. Therefore, growing AC use increases electricity consumption and impacts climate change. If the energy source is not renewable, it contributes to the urban heat island effect and ambient heat exposure [7]. There are numerous studies about AC "energy-use (EU)" in residential and educational buildings. AC is a factor that has a significant positive effect on the increase in household and educational building's electricity consumption [8–10]. In households in Asia, energy consumption increases with

the popularity of air conditioners [11]. Furthermore, studies about indoor thermal comfort in school classrooms have also been conducted. The air quality and temperatures in classrooms are essential factors in the learning process and improving them should be highly prioritized [9,12]. One study indicated that some classrooms in the UK had experienced overheating for more than 40% of school hours [13]. It found that indoor climatic conditions, measured during a field study in naturally ventilated classrooms in Tokyo and Yokohama, did not fall within the summer comfort thermal environments set by "The American Society of Heating, Refrigerating and Air-conditioning Engineers (ASHRAE)" 55–92. The conditions cannot possibly please everyone's boundaries, although, as expected, air-conditioned classrooms did feel well within the comfort zone boundaries [14]. Based on the author's previous research, when the daily average outside air temperature is between 29 °C and 30 °C, the AC is turned on with a 24 °C setting temperature, and the indoor air temperature in the classroom does not exceed the school hygiene standards of 28 °C, and there is no significant risk related to heatstroke [15]. Another study also found that the occupants of the classrooms could concentrate on studying more than before the introduction of AC and showed a positive view toward installing AC in classrooms [16].

Various local governments in Japan have used the PFI method for AC equipment maintenance projects in elementary and junior high schools. The PFI method only managed the energy of the AC equipment for heating and cooling, while other energy consumption management had been conducted before the PFI method was introduced in the schools. PFI is a method to provide efficient and effective public services by utilizing private funds and know-how for the design, construction, maintenance, and operation of public facilities and providing public services under the private sector's initiative [17]. It found that the AC EU was lower in schools where the PFI method was adopted than in schools where the conventional or lease method was adopted [18]. Many local governments that have introduced AC equipment through the PFI project are monitoring the AC equipment performance by installing measuring instruments and collecting data. Oita City is an example of a municipality that is introducing AC equipment using the PFI method. By September 2019, the AC equipment installation rate for ordinary classrooms in public elementary and junior high schools in Oita City reached 100% [4]. However, the AC equipment installation using the PFI method in Oita Junior High School, as a target building, was completed years after the school was built, and the school building was not planned with space designed for AC equipment. Two pieces of AC equipment were later installed in each class under the ceiling next to the window. Thus, it is predicted that there will be air temperature distribution differences based on position in the classroom.

In this research, the AC EU data monitoring from one year will be evaluated to determine the amount of EU after AC equipment is installed with the PFI method. In addition, the impact of the AC operating time and AC setting temperature will also be considered when determining the EU. Furthermore, the government set a recommended temperature for the AC setting, which is 28 °C in the summer (cooling) and 20 °C in the winter (heating), to optimize the energy-saving strategy [19]. Therefore, this research will investigate the AC setting temperature to determine if the recommended AC setting temperature value is retained. In addition to analyzing the AC EU, this paper will also evaluate the indoor thermal comfort in typical classrooms by position and zone. Zones will be divided into three: the "perimeter zone (PER)", which is near windows; the "center zone (CNT)"; and the "interior zone (INT)", which is near the corridor. It is necessary to examine these zones based on a previous study that states overheating occurs due to solar gains through large windows, as the result of providing daylight in classrooms, high levels of thermal insulation, and air sealing the building envelope, resulting in discomfort and reducing student performance [20]. This research result is hoped to be a reference for AC energy-saving strategies, AC layout installation, and seating positions to optimize indoor thermal comfort. In addition, the results of this research are further hoped to contribute to low-carbon building design technology development to attain a sustainable urban city.

Table 1 shows the abbreviation list with each meaning.

**Table 1.** Abbreviation list.

| Abbreviation | Meaning |
|---|---|
| AC | air-conditioning |
| AFSV | air flow sensation vote |
| ASHP | air source heat pump |
| ASHRAE | The American Society of Heating, Refrigerating and Air-conditioning Engineers |
| AT | air temperature |
| BESCS | Building Environmental Sanitation Control Standards |
| CNT | center zone |
| COP | coefficient of performance |
| EHP | electric heat pump |
| EU | energy-use |
| HVAC | heating, ventilation, and air-conditioning |
| INT | interior zone |
| ISO | International Organization for Standardization |
| JSEHMS | Japan School Environmental Hygiene Management Standard |
| LPG | liquefied petroleum gas |
| MEXT | Ministry of Education, Sports, Science, and Technology, Japan |
| PER | perimeter zone |
| PFI | Private Finance Initiative |
| PMV | predicted mean vote |
| REHVA | Federation of European Heating, Ventilation and Air Conditioning Associations |
| RH | relative humidity |
| TSV | thermal sensation vote |

## 2. Methods

The research framework is shown in Figure 1. This study examines interactive relationships among three methods: experiment or actual measurement, questionnaires, and PFI monitoring data. This research used sensitivity analysis to compare those three research methods. AC EU will be examined by analyzing PFI monitoring data obtained from Oita's municipal office. All PFI data obtained from Oita's municipal office include only the AC management data for cooling and heating. The monitoring data analysis period for the AC EU analysis is from April 2019 to March 2022 since Japan's school academic year starts in April. In addition to the AC EU, the AC operating times and AC setting temperature will be analyzed with sensitivity analysis using the PFI monitoring data. The AC operating times are calculated as the average per room, while the AC EU is the total energy use in all classes. The measurement of the AC EU with the PFI method is conducted with an internal system installed in each piece of AC equipment from the AC purchase plan stage for monitoring implementation, and all data will be collected. In addition, the system also measures suction air temperature returning to the AC equipment for air-conditioning control. The AC indoor unit suction temperature utilizes the temperature output from the built-in thermistor to the AC indoor unit inlet (suction port) for the air-conditioning control system. In this research, this suction air temperature, which is positioned at AC level (2,8 m), will be called "Air temperature with PFI". This suction air temperature monitoring data will be used for the indoor thermal comfort analysis compared to the actual air temperature measurement and PFI monitoring data results. Figure 2 shows the inside a typical classroom of Oji Oita Junior High School (Figure 2a) and the position of the measurement item, TR-72NW (Figure 2b).

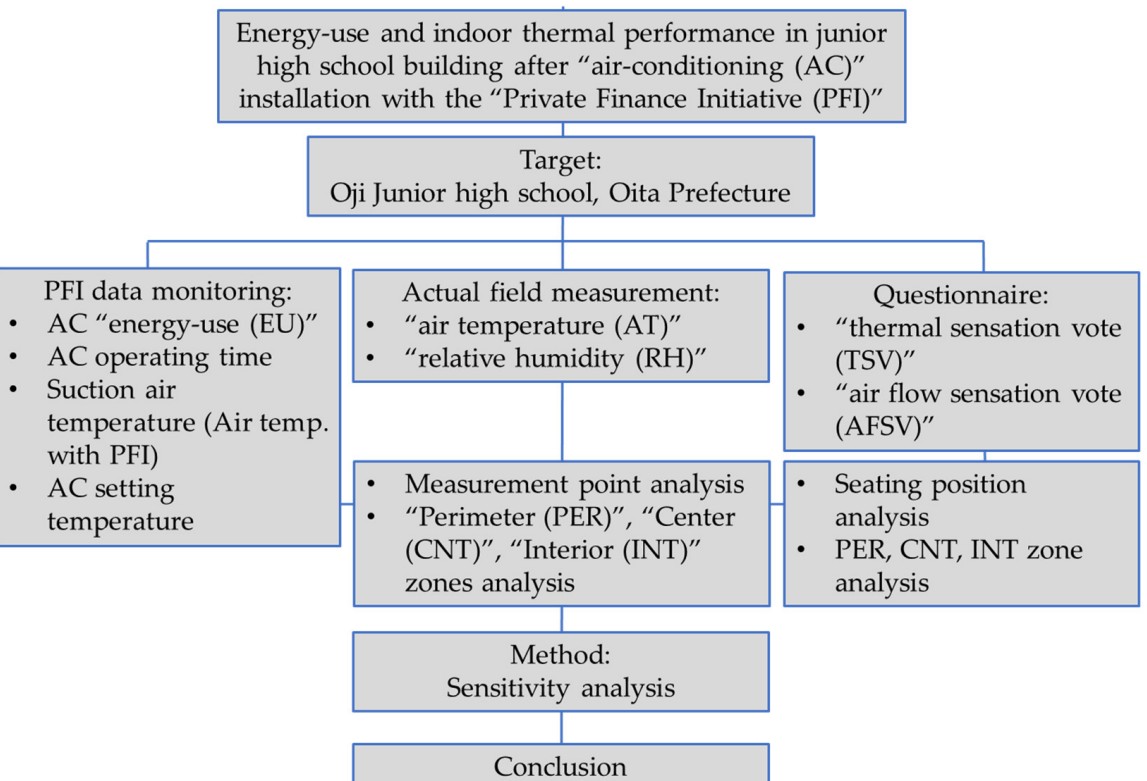

**Figure 1.** Research framework.

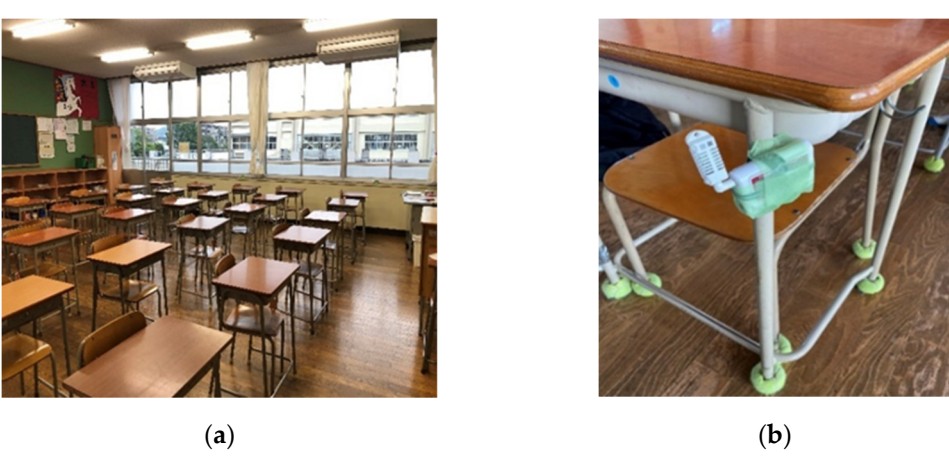

| (**a**) | (**b**) |

**Figure 2.** Field measurement condition (**a**) Inside a typical classroom of Oji Oita Junior High School and (**b**) the position of the measurement item, TR-72NW.

This paper will evaluate indoor thermal performance in each point of measurement and zone (Figure 3) using field measurements. The measurement item used in this research was the air temperature and relative humidity recorder, TR-72NW (Figure 2b), with a measurement range of 0~55 °C, 10~95%RH, and an accuracy of ±0.5 °C, ±5%RH (at 25 °C, 50%RH) [21]. The actual measurements were conducted in the summer of 2019 in Oji Junior High School's air-conditioned classrooms located in Oita City, Japan. The school's number of typical classrooms in 2019 was 20, and in 2020, the number was 21, while the total number of students in 2019 was 558 students, and in 2020, the total number was 583 students [22,23]. The actual thermal sensation and airflow sensation questionnaires were also distributed to derive the subjective evaluation of the thermal comfort of students

in each seating position. The thermal sensation vote questionnaire value complies with the PMV method value on a discrete seven-point scale by ASHRAE [24].

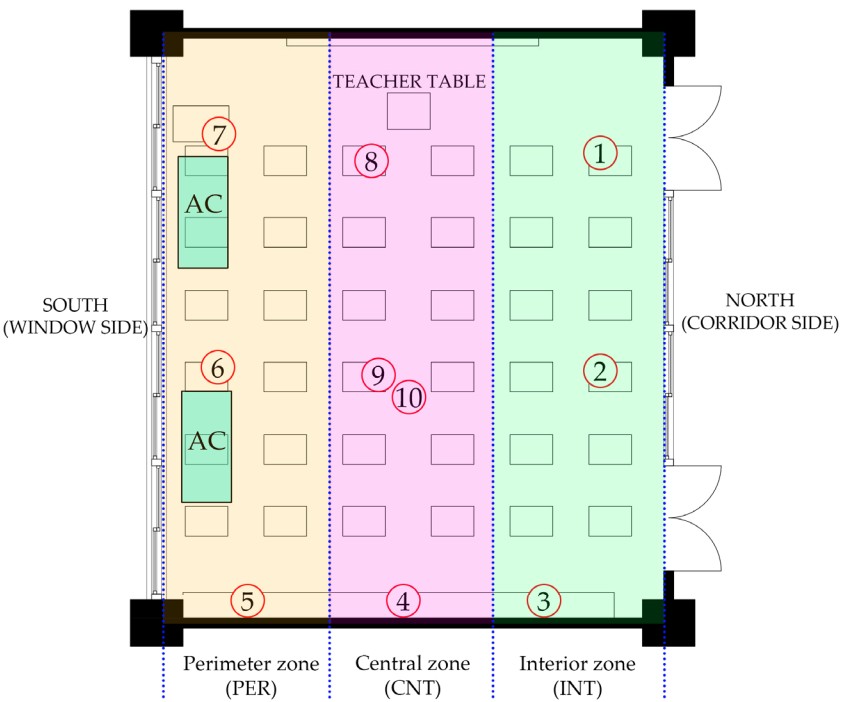

**Figure 3.** Measurement points and zoning.

The indoor thermal comfort analysis will be divided into four parts including field measurement results, zone correlation, a comparison between the field measurement result and PFI monitoring data result, and a questionnaire. Figure 3 shows the measurement points and zoning in each classroom. The plan was divided into three zones: PER, CNT, and INT. Points ①, ②, and ③ represent the INT. CNT is represented by points ⑧, ⑨, and ④, while points ⑦, ⑥, and ⑤ represent PER. The measurement height of points ①, ②, ⑥, ⑦, ⑧, and ⑨ are 70 cm, point ⑩ is 10 cm, while points ③, ④, and ⑤ are 100 cm. The interval of this measurement was 10 min. Point ⑩ is measured to assess the thermal comfort in the classrooms and whether it causes temperature stratification.

The measurement and questionnaire period are shown in Table 2. The measurement time for the analysis was data from 08:00 to 16:00. The analysis target period was only 4 days, considering the weather, where sunny days were chosen, and the questionnaire was distributed on only one day.

**Table 2.** Measurement and questionnaire distribution period.

| Actual Measurement | | Questionnaire Period |
|---|---|---|
| **Measurement Period** | **Analysis Target Period** | |
| 27 August 2019–9 September 2019 | 4–6 September 2019, 9 September 2019 | 4 September 2019 |

The type of AC equipment used in Oji Junior High School is an "air source heat pump (ASHP)". Oji Junior High School installed an "electric heat pump (EHP)" and "liquefied petroleum gas (LPG)" system for the AC system. AC EU, in power consumption (kWh) and gas (LPG) consumption (m$^3$), was determined using the PFI method that collected only AC EU and not any other energy use. AC EU is calculated from the primary data obtained in kWh (electric) and m$^3$ (gas) and converted to GJ with each heat source conversion (unit calorific value), which are shown in Table 3. The formula to convert power consumption to

energy consumption (crude oil equivalent) is kWh × unit calorific value. The Enforcement Regulations of the Law Concerning the Rational Use of Energy stipulate the numerical value for converting electric power into energy consumption. The unit calorific value for daytime electricity is 9.97 MJ/kWh [25].

**Table 3.** Unit calorific value [25].

| "Electric Heat Pump (EHP)" | "Liquefied Petroleum Gas (LPG)" |
| --- | --- |
| 9.97 MJ/kWh | 100.47 MJ/m$^3$ |

## 3. Results

### 3.1. Oita City Climate

The climate of Oita Prefecture, located in the northeastern part of the main island of Kyushu, Japan, generally belongs to the warm and temperate summer rain type heavy rain climate. Oita City, the target school location in the central part of Oita Prefecture, has 1800 mm or less annual precipitation [26]. The weather in the winter is relatively good [27]. Figure 4 shows the monthly average air temperature and relative humidity in Oita City from April 2019 to March 2022. The peak of the summer was in August at 27.1~27.2 °C, while the peak of the winter 2020 and 2021 was in January at 6.7~7.7 °C, and the peak of the winter 2022 was in February at 5.9 °C. Based on the monthly climate parameters in Oita City, schools start to use AC regularly from June to September in the summer and from December to March in the winter.

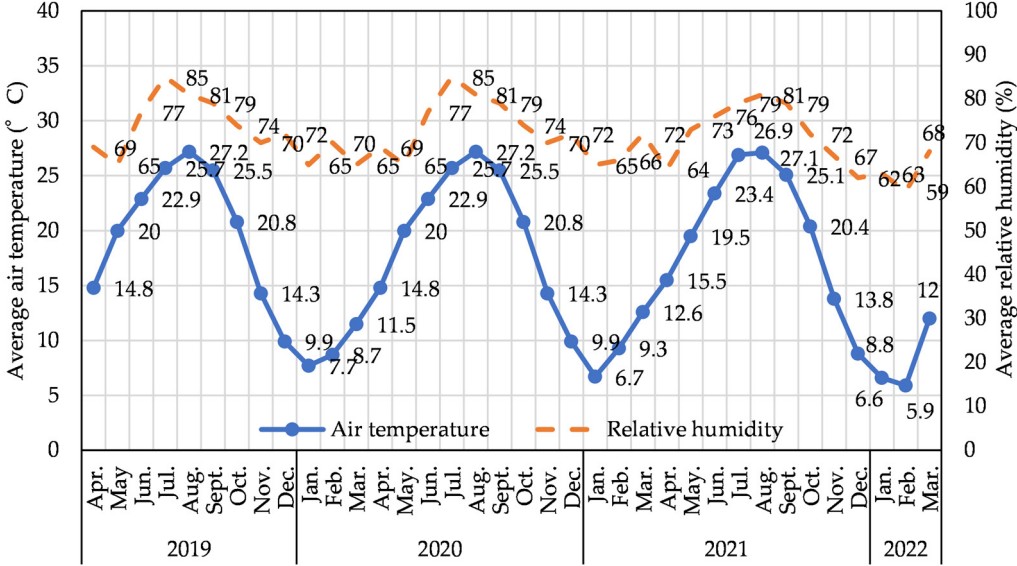

**Figure 4.** Oita City April 2019–March 2022 monthly air temperature and relative humidity.

### 3.2. AC Energy-Use Data Result

In this research, AC EU is analyzed by totaling the data, and AC operating time is analyzed by averaging the data per room. Based on the school's yearly total AC EU and the average AC operating time per room from April 2019 to March 2022 (Figure 5), the AC frequently operates from December to March in the winter season and July to September in the summer. Figure 5 shows that the AC EU in September 2019 (61.1 GJ) was 1.6 times the AC EU in July 2019 (37.8 GJ) and 3.1 times the AC EU in August 2019 (19.6 GJ). The AC EU in August was not high, considering the summer holiday in that month. There was an extreme escalation of the AC EU in the 2020–2021 and 2021–2022 data, considering the COVID-19 pandemic outbreak that happened during that time, which led to high AC EU due to a combination of AC and natural ventilation.

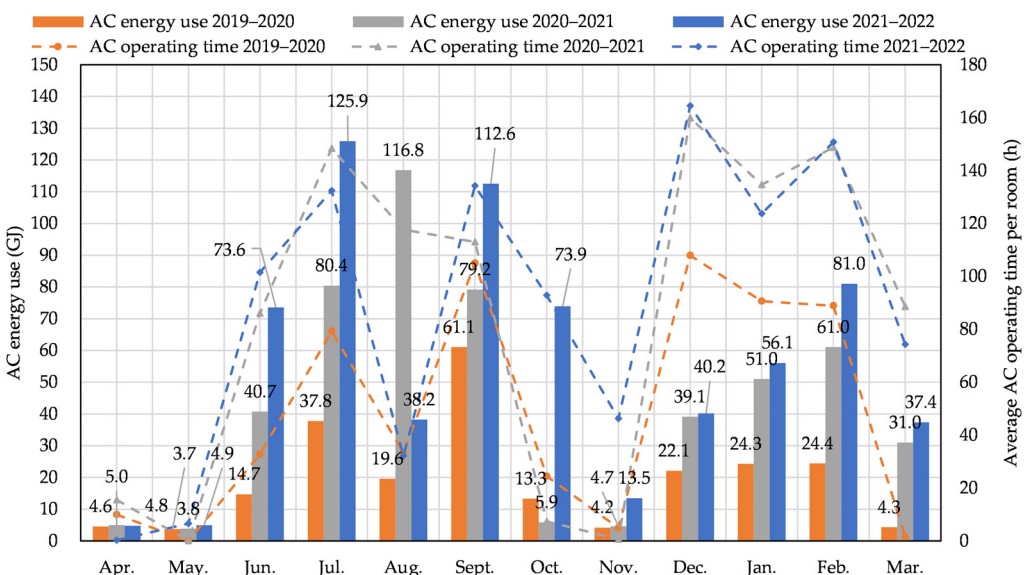

**Figure 5.** Oji Junior High School's AC EU and AC operating time (2019–2022).

In August 2020, the AC EU was irregularly high compared to June, July, and September. This was caused by the summer holiday abolishment in August to substitute lockdown and online lessons in March 2020 due to the pandemic outbreak [9]. The AC EU in June to October 2021 was highly escalated compared to 2020. The AC EU escalation in October 2021 was caused by longer AC operating times. The AC EU from June to September 2021 was higher than in 2020, even though the AC operating times are not significantly different. Figure 6 shows that this occurred because the AC setting temperature in the summer of 2021 was lower than in 2020. Opening windows and doors to lower the $CO_2$ concentration levels below 1000 ppm to prevent virus transmission [28,29], as a school protocol, also affects the indoor thermal environment, which leads to a lower AC setting temperature in the summer. Based on the author's previous research result, the $CO_2$ concentration will be high and exceed 1000 ppm after 30 min when smoke exhaust windows are closed in the classroom of a discussion-type class with 419.9 m³ volume area and a total of 55 people [30].

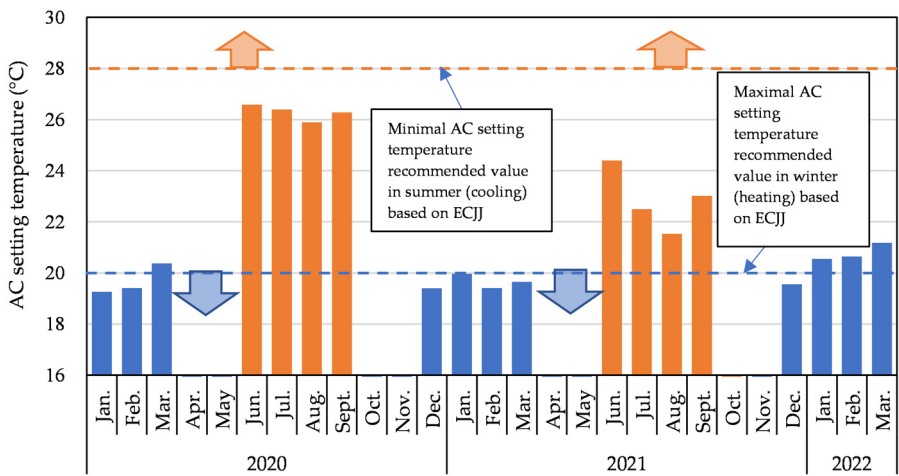

**Figure 6.** Oji Junior High School's AC setting temperature (2020–2022).

Even though the AC operating time from December 2019 to February 2020 was high, the AC EU in these months was not as high as in the summer. It can be presumed that the AC setting temperature greatly impacted this AC EU difference in the summer and winter. The recommended value for the AC setting temperature in classrooms, as given by the government, is 28 °C (cooling) in the summer and 20 °C (heating) in the winter

to optimize the energy-saving effect [19]. Nonetheless, the AC setting temperature in the summer may not meet the recommended value (Figure 6). The AC operating times in the winters of 2020–2021 and 2021–2022 were longer than the winter of 2019–2020 because of the COVID-19 pandemic, which requires longer AC operating times 2 h before and after occupancies as recommended by "Federation of European Heating, Ventilation and Air Conditioning Associations (REHVA)" and ASHRAE [31]. Figure 6 shows that in the winter season, except in January to March 2022, the AC temperature is set under 20 °C, which still meets to government AC setting temperature recommended value.

During winter, the Oita Prefecture has more sunshine hours than other Prefectures in Kyushu [32] and is known to have warmer weather in the winter than Northside Prefectures in Japan. This made the "coefficient of performance (COP)" of the heating mode in the winter higher than the cooling mode in the summer. During the heating mode, the higher the outdoor air and the lower the AC setting temperature, the higher COP. Similarly, during the cooling mode, the temperature lift is minimized, and COP is maximized if coldness is distributed at the warmest possible temperature and the heat is rejected at the lowest possible temperature [33]. The COP difference between the heating mode and the cooling mode in Oita could be the cause of the AC EU difference between the summer and winter even though the AC operating time in the winter was higher than in the summer. Another assumption is that internal heat generation, such as people's heat generation, in the winter has more impact on heating the indoors, which causes a lower AC setting temperature than the government-recommended value of the AC setting temperature at 20 °C.

### 3.3. Summer Indoor Thermal Comfort Result
3.3.1. Field Measurements Result

- "Air Temperature (AT)" measurement result

The analysis period of measurement is from September 4th to 9th. The outdoor air temperature and relative humidity are shown in Figure 7. Figures 8–10 show the air temperature data results for classes A, B, and C. All the data in points ⑥ and ⑦ (PER) were excluded because the sun radiation exposure on the measurement instruments caused extremely high temperatures. Based on the three graphs of data results, it can be seen that point ⑤ had the highest temperature, while point ⑧ had the lowest temperature. It can be assumed that point ⑤ was located near the window (PER), while point ⑧ was located in the center zone, coinciding with the position of the AC wind blow. Based on Figures 8–10, there were oscillatory behavior or hunting phenomena from day 1 to day 4 of the measurement period in each class. There were some small-range hunting phenomena and some wide-range hunting phenomena. The small-range hunting phenomena can be presumed to be because the door and window were occasionally opened. The wide-range hunting phenomena indicate that there were some discontinuations in AC use within these ranges. Hence, the outdoor air temperature affected the indoor air temperature, which caused AC adjustment due to the differences between indoor and outdoor air temperatures. This oscillatory behavior is determined to be one of the common HVAC system disturbances, which control the AC setting temperature and suction air temperature at about 1.5–2 °C. This may happen because of the outdoor temperature changes, or the occupancy of the rooms being conditioned [34].

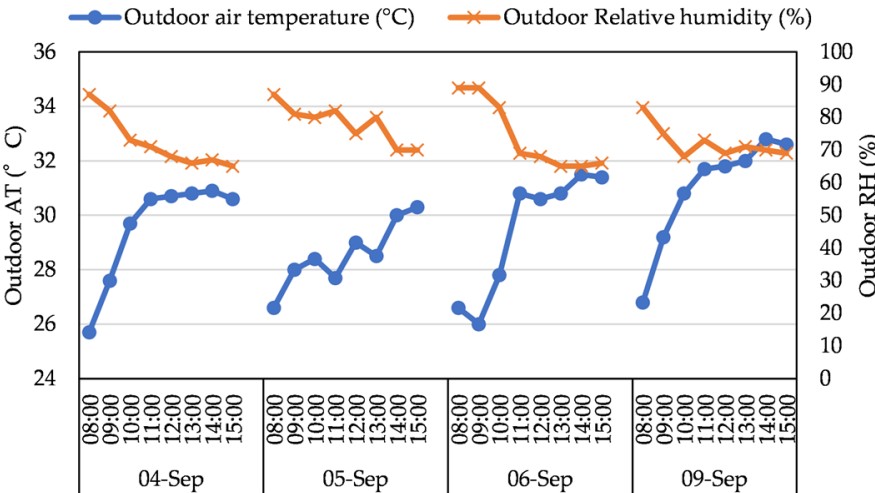

**Figure 7.** Outdoor summer air temperature and relative humidity in measurement time.

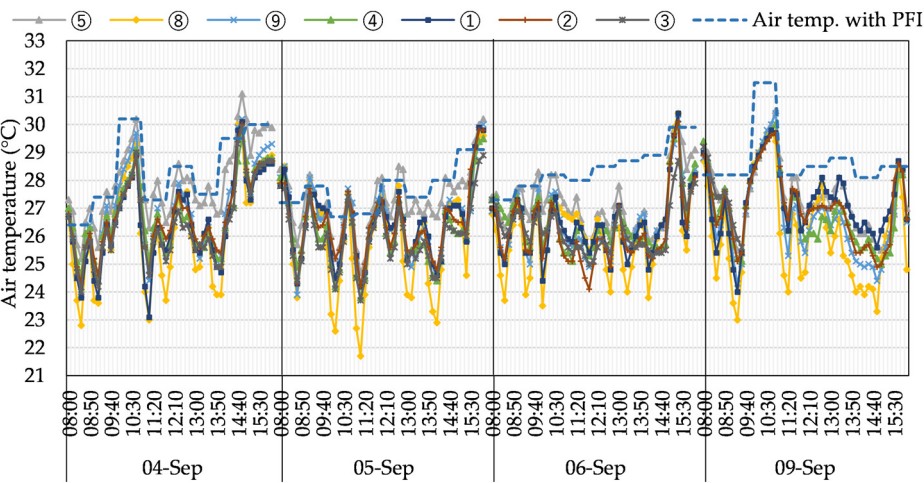

**Figure 8.** Summer air temperature for class A.

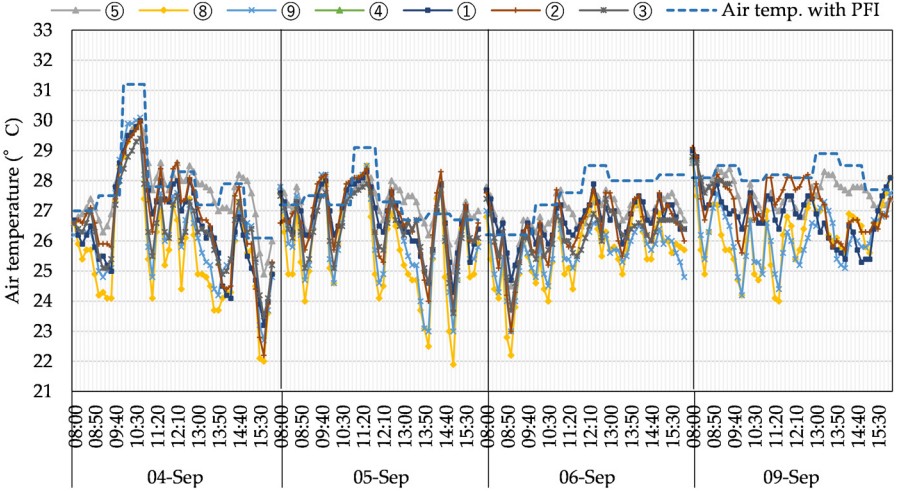

**Figure 9.** Summer air temperature for class B.

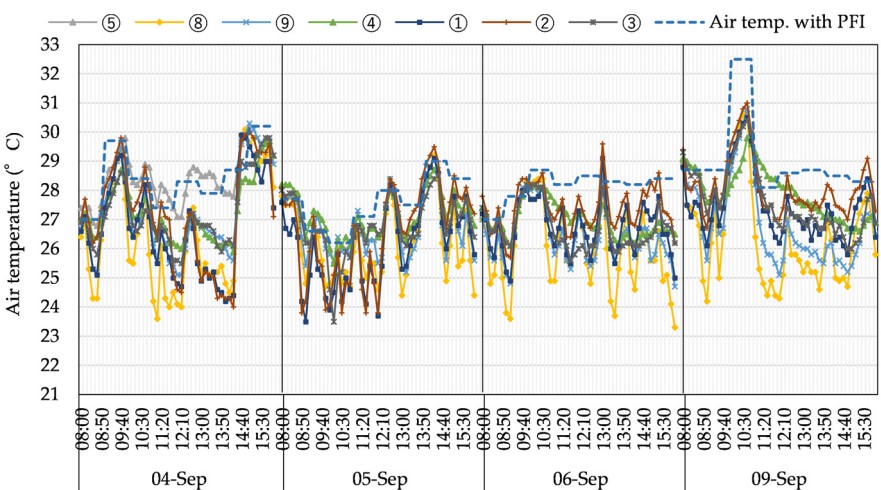

**Figure 10.** Summer air temperature for class C.

The air temperature result for class A (Figure 8) shows that the indoor air temperature in the classroom was unstable from time to time. Some points reach an air temperature above 30 °C, and point ⑧ sometimes reaches an air temperature below 23 °C. There are six periods of time when the air temperature reaches 28 °C to 30 °C within this measurement period. It can be predicted that the high air temperature was caused by turning off the AC.

Similar to the class A result, Figure 9 shows that the air temperature in class B was also unstable. However, there was only one period of time when the air temperature reached 28 °C to 30 °C within this measurement period.

Figure 10 shows the air temperature result for class C, which is more similar to the result for class A, in which the air temperature reached 28 °C to 30 °C about six times within this measurement period.

Based on MEXT, the comfortable temperature range in classrooms for all seasons is 18 °C to 28 °C [35,36]. However, in this research, the indoor thermal comfort ranges from 25 °C to 28 °C in the summer. This lower limit (25 °C) is based on the school environmental hygiene standards revision, which stated that the most desirable conditions for learning, which do not place a physical or psychological burden on students, are 18–20 °C in the winter and 25–28 °C in the summer [37]. The lower limit of the range above 18 °C in the summer is for the AC energy-saving strategy consideration. In this study, the "Predicted mean vote (PMV)", a standard method to measure indoor thermal comfort described by ASHRAE Standard 55, could not be calculated, which is a limitation of this research. This experiment did not measure other parameters, such as PMV or globe temperature and air velocity, due to the complex sensor installation and restraining teaching and learning activities intervention required to measure the other parameters. Therefore, the air temperature and relative humidity were measured using small-size measurement items, which did not interfere with students' activities. However, in this study, as previously mentioned, the air temperature range, 25 °C to 28 °C, is considered the comfort range and is based on the revision of the school's environmental hygiene standards [37]. Furthermore, it was also obtained from acceptable Predicted Mean Vote (PMV) by the "International Organization for Standardization (ISO)" 7730:2005 as range for existing buildings between −0.7 and +0.7 [38]. The calculation data assumed to determine the range standard with an acceptable PMV by the ISO are shown in Table 4.

**Table 4.** PMV calculation parameters and assumption values.

| PMV Calculation Parameters | Assumption Values |
|---|---|
| Metabolic rate | 1 met (58.2 W/m$^2$) |
| External work | 0.0 W/m$^2$ |
| Relative humidity | 60% (obtained from averaged RH measurement data) |
| Clothing insulation | 0.5 CLO |
| Air velocity | 0.2 m/s |
| Radiant temperature | Equal to air temperature |

The PMV calculation result [39–41] for a lower limit air temperature of 25 °C is PMV −0.65 with PPD 13.86%, and for an upper limit of 28 °C is PMV +0.6 with PPD 12.44%.

Figure 11 shows the air temperature percentage in each range. The total average air temperature percentage of each range is shown in Figure 11a. It shows that point ⑤ (PER zone) had the highest percentage of air temperature above 28 °C, 25.5%, while point ⑧ had the highest air temperature below 25 °C, 26.7%. Point ③ had the highest air temperature comfort range (25~28 °C), 84.2%. Similar to the total average result, the air temperature percentage (Figure 11b–d) shows that the highest comfort range of air temperature in each class was at point ③, and the smallest comfort range of air temperature was at point ⑧. Point ④ data in class B (Figure 11c) could not be acquired due to measurement error. Point ⑤ data in class C (Figure 11d), which was successfully obtained only on September 4th due to setting measurement error results, has an extremely high percentage of air temperature above 28 °C, 62.8%.

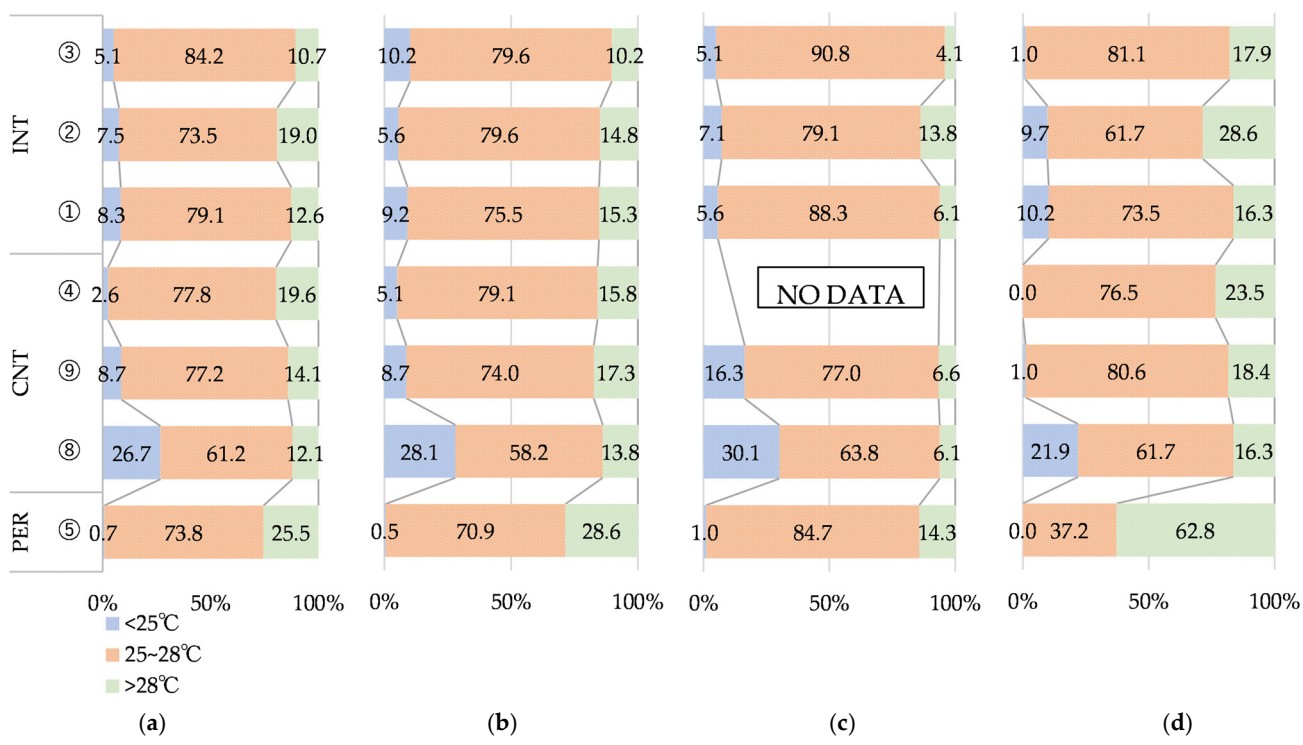

**Figure 11.** Summer air temperature comfort range percentage for the (**a**) total average and in (**b**) class A, (**c**) class B, and (**d**) class C.

- "Relative Humidity (RH)" measurement result

High humidity increases the risk of heat stroke, so paying attention to humidity is necessary. In this section, relative humidity is analyzed using psychometric charts. Figures 12–14 show the psychometric charts for each class. There are two standards for relative humidity. One is the "Japan School Environmental Hygiene Management Standard

(JSEHMS)", which ranges between 30 and 80% for an acceptable comfort relative humidity range in the classroom [37]. The other is the "Building Environmental Sanitation Control Standards (BESCS)", which ranges between 40 and 70% for an indoor acceptable comfort relative humidity range [36,37]. Figures 12–14 show that almost all data (above 90%) are within the JSEHMS comfortable range of 30 to 80%. However, in this psychometric chart analysis, the air temperature will also become a factor of the comfort range.

Although the air temperature comfort range is discussed previously in the air temperature analysis, in this analysis, the summer comfort range percentage will be analyzed with air temperature and relative humidity data using the psychometric chart.

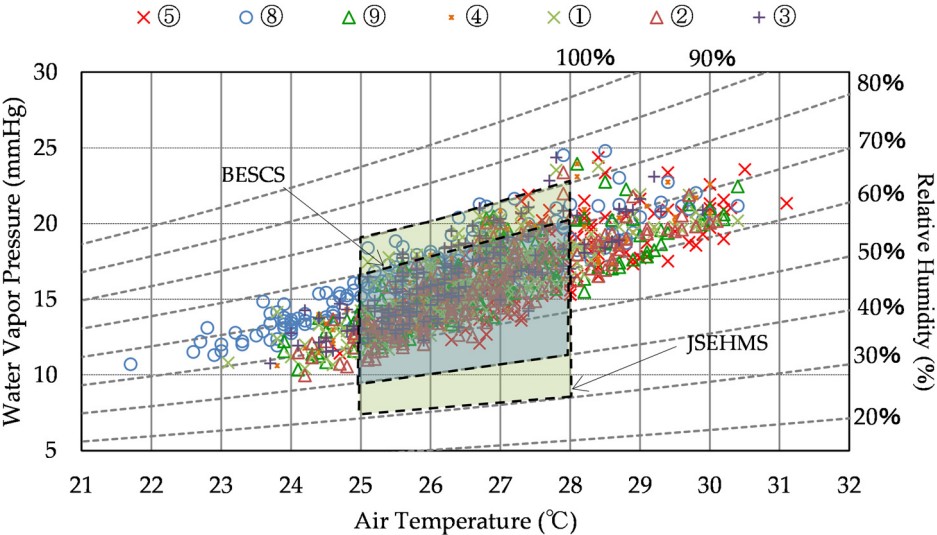

**Figure 12.** Summer psychometric chart for class A.

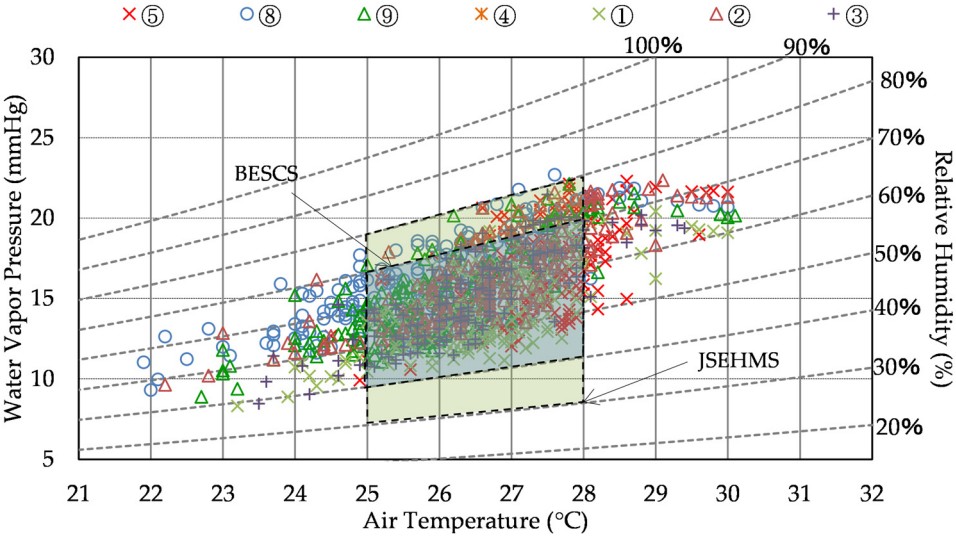

**Figure 13.** Summer psychometric chart for class B.

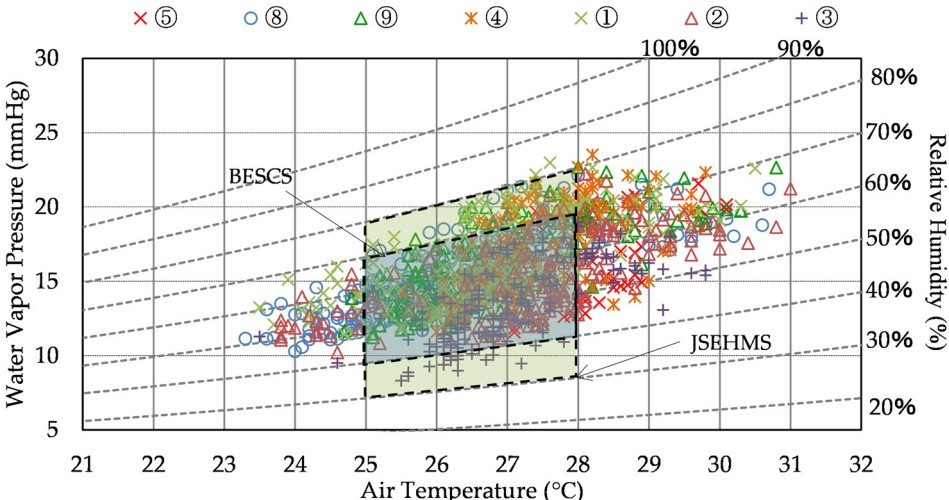

**Figure 14.** Summer psychometric chart for class C.

Figure 15 shows the comfort range percentage based on the JSEHMS, while Figure 16 is based on the BESCS. The summer comfort range result based on the JSEHMS (Figure 15) shows similarity to the summer air temperature comfortable range previously discussed since the relative humidity range is quite wide, from 30 to 80%. For the total average data, point ⑧ had the smallest percentage of relative humidity and air temperature comfort range based on the JSEHMS (Figure 15a) and BESCS (Figure 16a). On the other hand, point ③ had the highest comfort range percentage for the total average data based on the JSEHMS and BESCS. This likely happened since point ③ is near the door, which is occasionally opened so that the airflow from the door affects the relative humidity. Point ⑤ is the point in the perimeter zone and was above 70% comfort based on the JSEHMS (Figure 15b,c) and above 65% comfort based on the BESCS (Figure 16b,c) except for class C (Figures 15d and 16d). Data for point ⑤ in class C supplied in the graph is only from September 4th. However, it will not be further analyzed due to the lack of data on other measurement days. Meanwhile, data point ④ for class B is not supplied because no data were obtained on any measurement days due to measurement error.

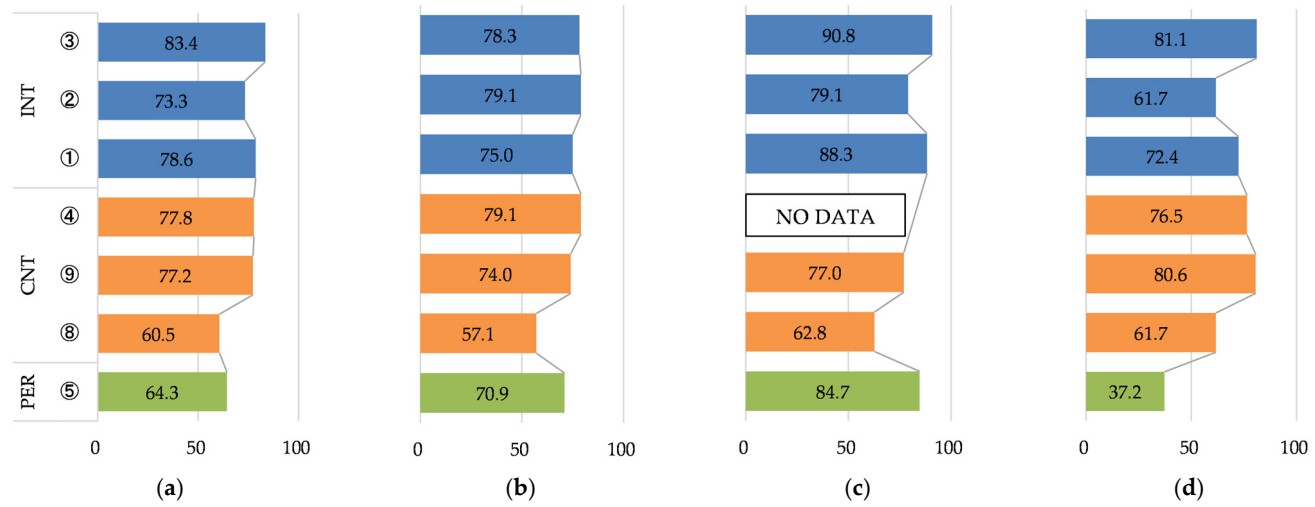

**Figure 15.** Summer comfort range percentage based on the JSEHMS for the (**a**) total average and in (**b**) class A, (**c**) class B, and (**d**) class C.

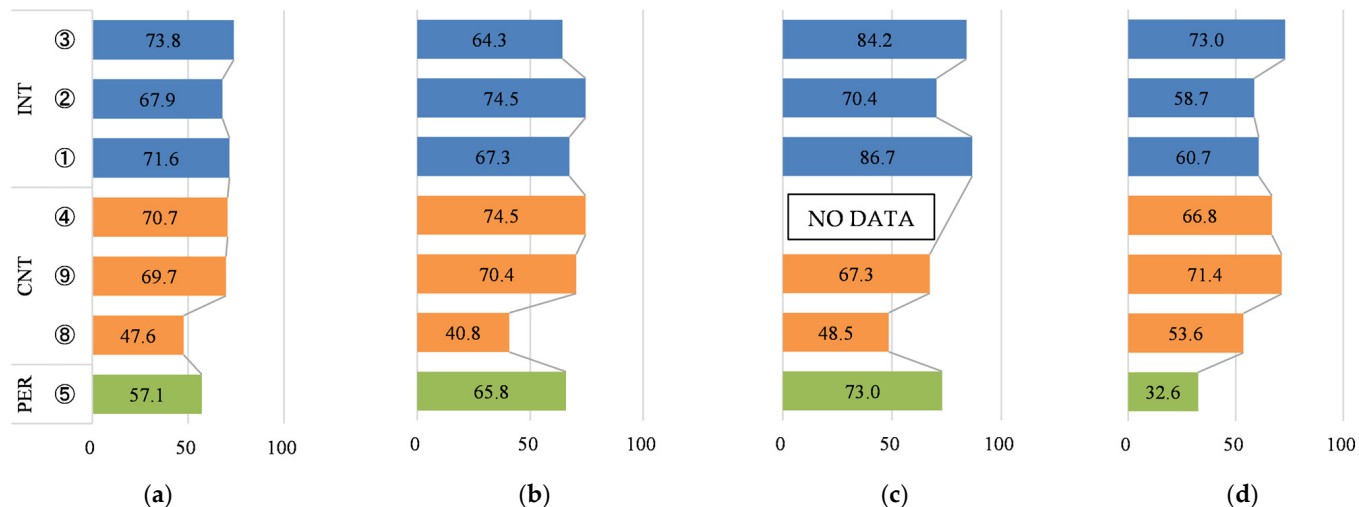

**Figure 16.** Summer comfort range percentage based on the BESCS for the (**a**)total average and in (**b**) class A, (**c**) class B, and (**d**) class C.

3.3.2. Perimeter, Center, and Interior Zone Correlation

The air temperature comparison between the PER, CNT, and INT zones in the summer is shown in Figures 17a, 18a and 19a. There is a limitation in this analysis since points ⑥ and ⑦, which are the points in the PER zone, were directly affected by direct solar radiation; thus, point ⑤ is the only point in the PER zone that is analyzed. Based on Figure 17a, the air temperature in the PER zone was higher than in the INT zone. Figure 18a shows that the air temperature in the PER zone was higher than in the CNT zone, while Figure 19a shows that the air temperature in the INT zone was higher than in the CNT zone. This indicates that air temperature in the PER zone was affected by solar radiation that came through the window.

Meanwhile, the CNT zone had the lowest temperature caused by the AC posi-tion in-stalled above the PER zone which blew the wind directly on the CNT zone. Figures 17b, 18b and 19b show the relative humidity correlation between the PER, CNT, and INT zones in the summer. The Figures show no significant difference in relative humid-ity between the PER, CNT, and INT zones, yet Figure 19b shows that the relative humidity in the CNT zone was slightly higher than in the INT zone. Therefore, it can be assumed that the doors were opened occasionally, which decreased the humidity in the INT zone.

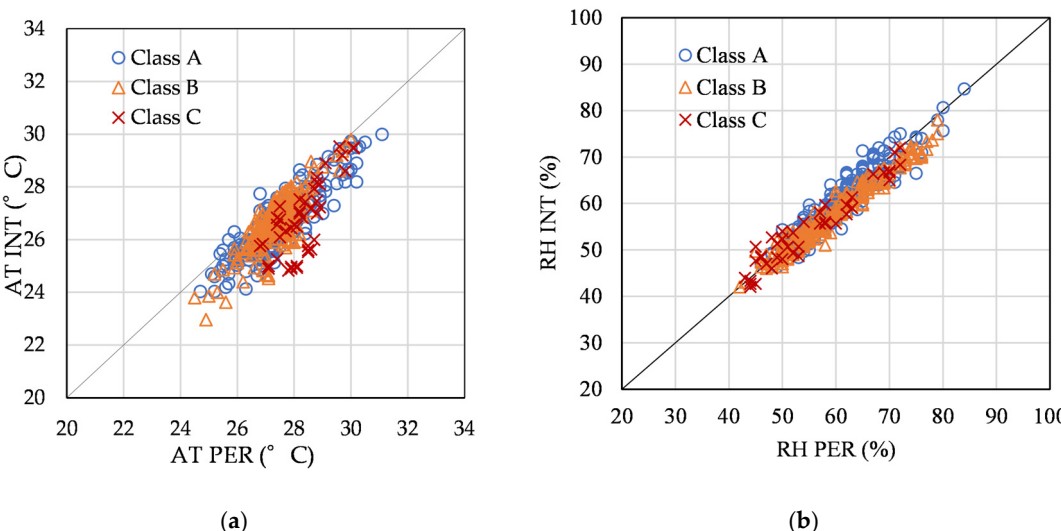

**Figure 17.** PER and INT correlation for (**a**) air temperature and (**b**) relative humidity.

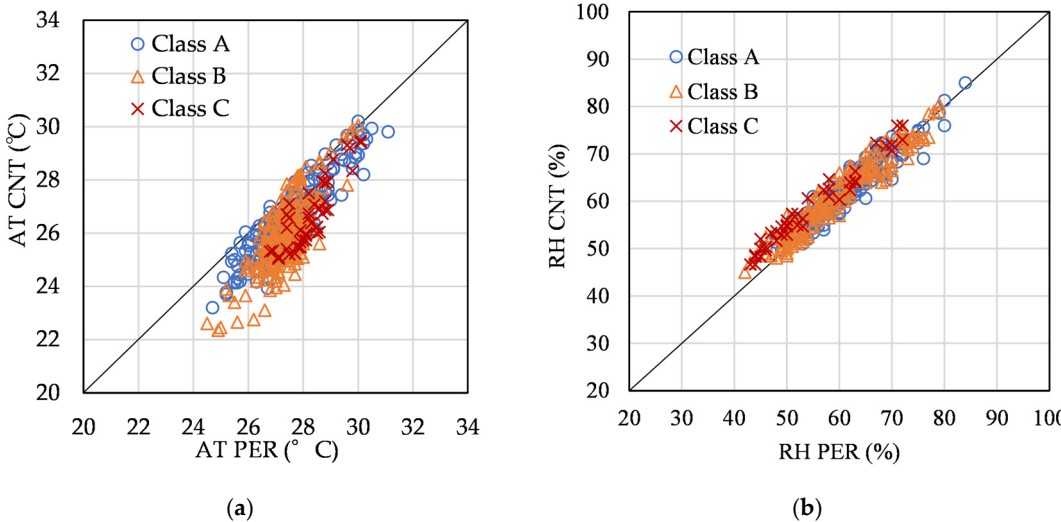

**Figure 18.** PER and CNT correlation for (**a**) air temperature and (**b**) relative humidity.

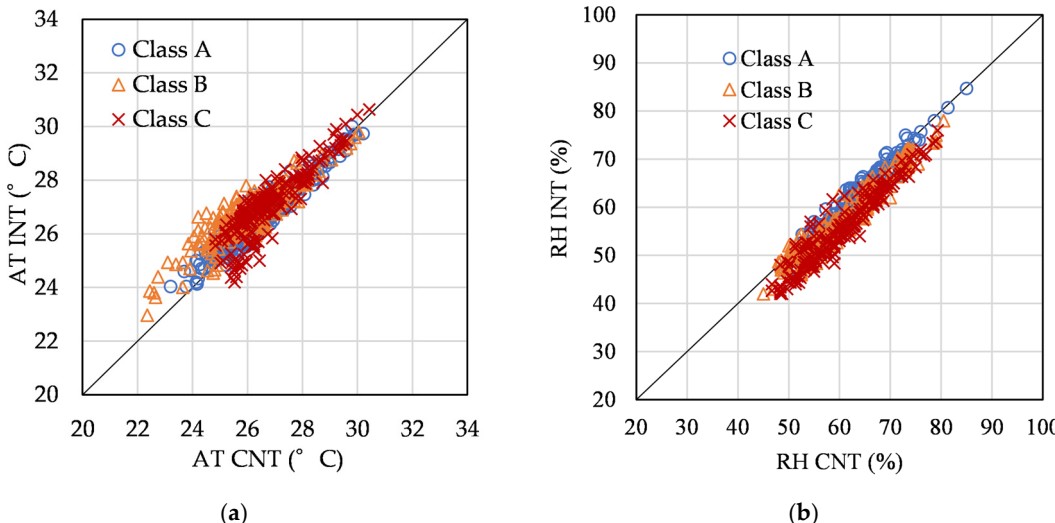

**Figure 19.** CNT and INT correlation for (**a**) air temperature and (**b**) relative humidity.

The Pearson correlation of the PER-CNT-INT zone is shown in Table 5. The air temperature Pearson correlations were on average under 0.9 except for class A in the PER-CNT and CNT-INT zone, while all relative humidity Pearson correlations were above 0.9. This indicates typical air temperature differences between the zones.

**Table 5.** Pearson correlation of the PER-CNT-INT zone.

| | PER-INT | | | PER-CNT | | | CNT-INT | | |
|---|---|---|---|---|---|---|---|---|---|
| | **Class A** | **Class B** | **Class C** | **Class A** | **Class B** | **Class C** | **Class A** | **Class B** | **Class C** |
| Air temperature | 0.888 | 0.829 | 0.754 | 0.909 | 0.775 | 0.857 | 0.955 | 0.886 | 0.891 |
| Relative humidity | 0.933 | 0.976 | 0.959 | 0.948 | 0.963 | 0.980 | 0.973 | 0.966 | 0.957 |

### 3.3.3. Field Measurement and PFI Monitoring Data Comparation

As monitoring data for indoor air temperature in classrooms, air temperature data is collected with the PFI method, which is the suction temperature returning to the AC equipment for air-conditioning control. The AC indoor unit suction temperature utilizes the temperature output from the built-in thermistor to the AC indoor unit inlet (suction port) for the air-conditioning control system. This suction temperature sensor is placed

in each AC indoor unit at a level about 2.8 m from the floor. In this analysis, this suction temperature is named "air temperature with PFI".

Figures 20a, 21a and 22a show the average air temperature, air temperature data collected with the PFI method, and the AC setting temperature in the summer. Since the air temperature data collected with the PFI method and the AC setting temperature data were hourly, the measurement data results, which had 10 min intervals, were averaged to hourly data in this analysis. The AC setting temperature in the classrooms mostly were not kept at the government-recommended value of 28 °C in the summer to optimize the energy-saving effect [19] because the AC setting temperature in the summer was mostly under 28 °C in each class based on the monitoring data result. This lower AC setting temperature is strongly assumed to cause high AC EU in summer.

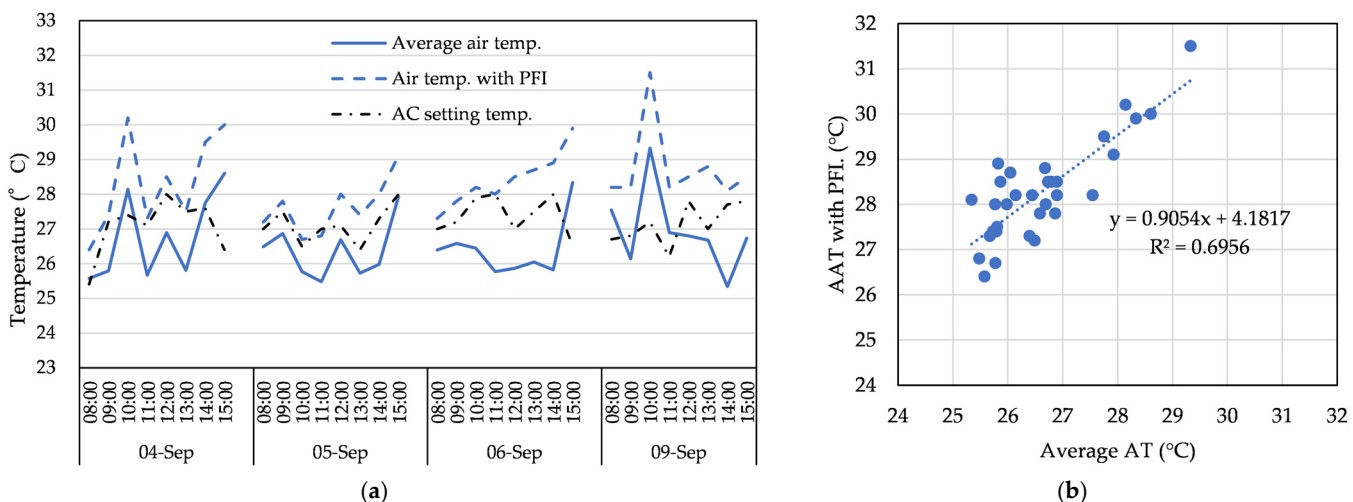

**Figure 20.** Summer indoor air temperature and correlation for the (**a**) summer average AT, AT with PFI, AC setting temp. of class A and (**b**) summer AT, AT with PFI correlation of class A.

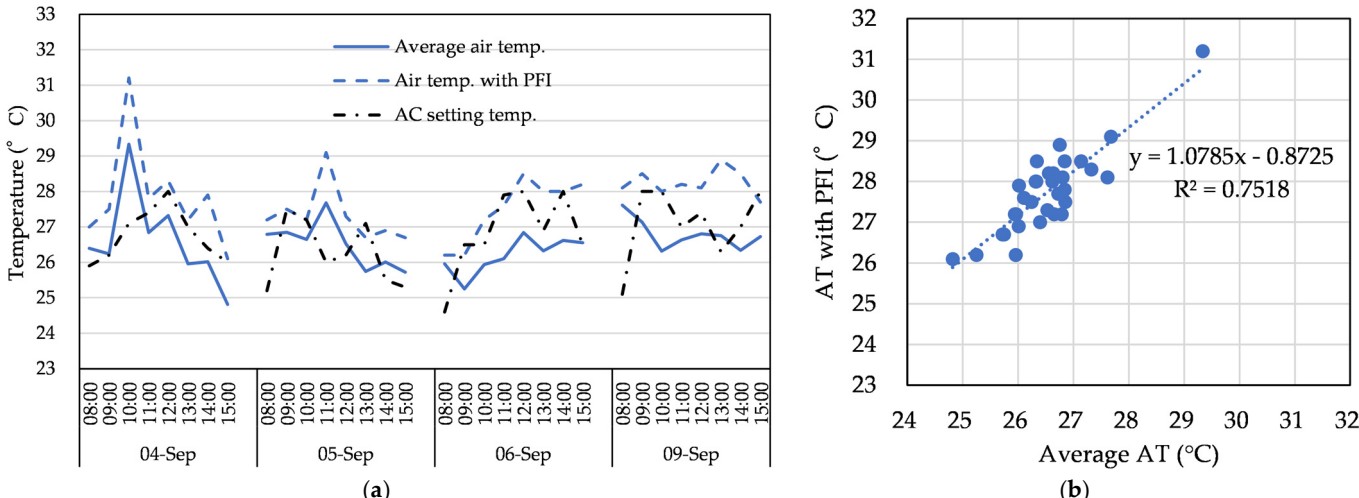

**Figure 21.** Summer indoor air temperature and correlation for the (**a**) summer average AT, AT with PFI, AC setting temp. of class A and (**b**) summer AT, AT with PFI correlation of class B.

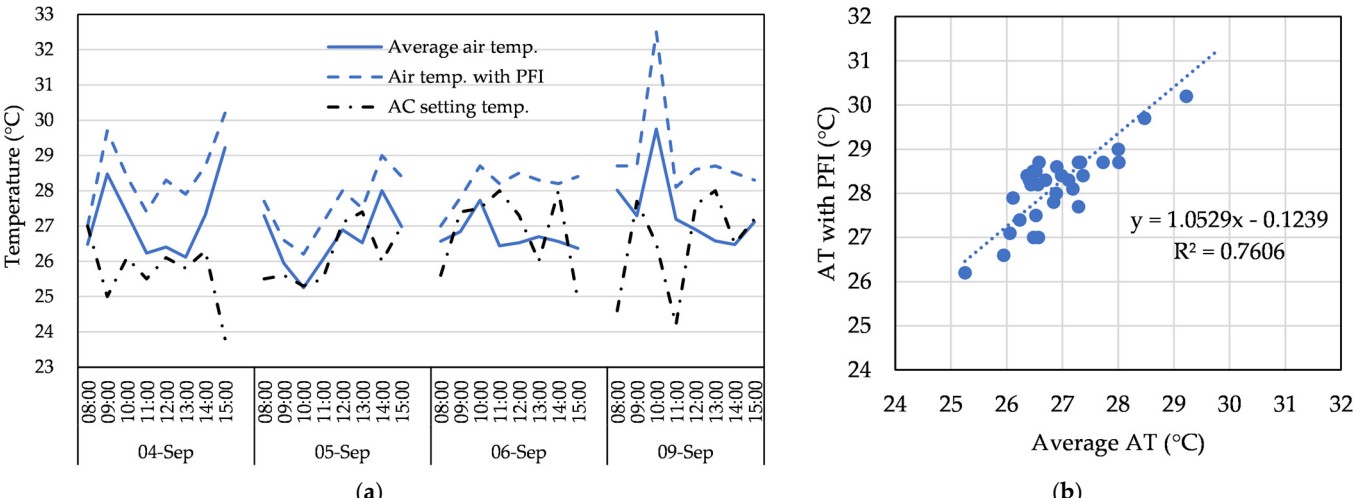

**Figure 22.** Summer indoor air temperature and correlation for the (**a**) summer average AT, AT with PFI, AC setting temp. of class A and (**b**) summer AT, AT with PFI correlation of class C.

The air temperature data collected with the PFI method (AC suction air temperature) were higher than the air temperature data result caused by the position of data reader sensors in the AC equipment, which were 2.8 m high. The AC setting temperature data in each classroom was unstable and related to the air temperature measurement data. Figures 20b, 21b and 22b show the correlation of the average air temperature and air temperature data collected with PFI. The simple regression equations were y = 0.9054x + 4.1817, $R^2$ = 0.6956 for class A, y = 1.0785x − 0.8725, $R^2$ = 0.7518 for class B, and y = 1.0529x − 0.1239, $R^2$ = 0.7606 for class C. Since the $R^2$ value was above 0.6, it can be said that there is a high correlation between the measured temperature and the air temperature data collected with the PFI method.

Figure 23a shows the outdoor and indoor air temperature differences in each classroom. The difference is from −1.2 °C to 7.5 °C, with a higher difference mostly reached after 11:00 to 15:00. September 9th had the highest outdoor–indoor air temperature difference due to a high outdoor temperature on September 9 compared with the other days (Figure 7).

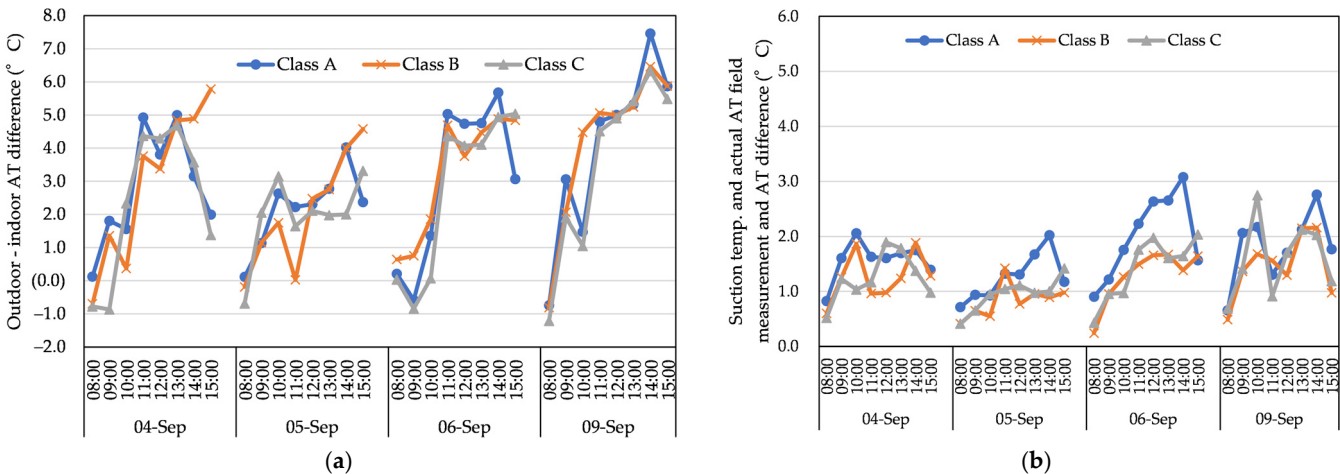

**Figure 23.** AT differences. (**a**) AT difference between outdoor and indoor AT. (**b**) AT difference between actual air temperature field measurement data and air temperature data collected with the PFI method in the summer season.

Figure 23b shows that the air temperature difference between the air temperature actual field measurement data and the air temperature data collected with PFI is about 0.2 °C to 3.1 °C. The air temperature actual field measurement data are the air temperature

average data of points ①, ②, ③, ④, ⑤, ⑧, and ⑨, with 70 cm and 100 cm height of the measurement items, while the air temperature data collected with the PFI method is the suction temperature in the AC level height (2.8 m). On the other hand, Figure 24 shows the air temperature distribution by the measurement levels of 10 cm, 70 cm, and 280 cm. The 10 cm measurement level is obtained from point ⑩ data, and the 70 cm measurement from point ⑨ data, which is located in the middle of the room and might be considered representative of average room air temperature. The air temperature data shown in the graph are averaged data from September 4 to 9 at 08:00–16:00. However, data for the 10 cm level in class C are the average air temperature data from September 9, 12:20, to the end of the measurement. This arose because the measurement item in point ⑩ in class C had failed to measure on September 9 from the beginning to 12:10. Based on the air temperature distribution result, the difference between the air temperature at the 70 cm level and the 10 cm level did not exceed more than 1 °C. Therefore, it can be stated that there is no extreme temperature stratification. However, the average air temperature difference between the 280 cm level and the 70 cm level exceeds more than 1 °C. This is not only caused by the high position but also caused by the position of the AC above the southern windows, which is the warmest side in the room, and caused by solar radiation effects. The air temperature difference between the suction air temperature at the 280 cm level and the room temperature at the 70 cm level does not necessarily affect the thermal comfort of the high position of the suction measurement level, which is not the level of the learning activities. However, this AC suction air temperature is also used for indoor air temperature data monitoring. Therefore, if indoor thermal monitoring in schools with the PFI method without actual measurement is conducted, this temperature difference between the PFI air temperature monitoring data and the actual field measurement results must be considered.

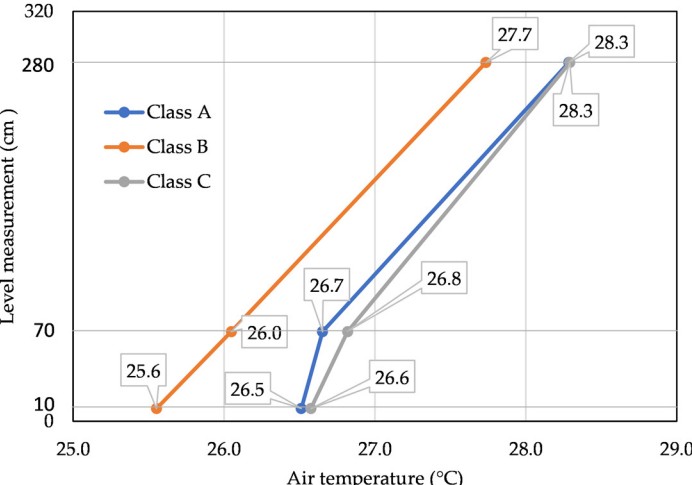

**Figure 24.** Average air temperature distribution per level measurement (10 cm, 70 cm, and 280 cm) in each classroom.

### 3.3.4. Questionnaire Result

The TSV and "air flow sensation vote (AFSV)" questionnaires were distributed on 4 September 2019. In this research, September could represent the entire summer period because the rainy season lasts from the beginning of June to mid-July, while August is the summer holiday. Furthermore, when the questionnaire was conducted, the students were asked to fill out the questionnaire as they generally felt about their indoor thermal sensation in the last seven days prior to 4 September 2019. In addition, the outdoor air temperature on this day reached above 30 °C after 11:00 (Figure 7), which fitted to the standard of the summer climate. Therefore, it could represent a sunny day at the peak of summer. The TSV and AFSV scales and definitions are shown in Table 6.

**Table 6.** Questionnaire scales and definitions.

| "Thermal Sensation Vote (TSV)" Scale | Definition | "Air Flow Sensation Vote (AFSV)" Scale | Definition |
|:---:|:---:|:---:|:---:|
| 3 | hot | 3 | much too still |
| 2 | warm | 2 | too still |
| 1 | slightly warm | 1 | slightly still |
| 0 | neutral | 0 | just right |
| −1 | slightly cool | −1 | slightly breezy |
| −2 | cool | −2 | too breezy |
| −3 | cold | −3 | much too breezy |

Figure 25 shows the summer TSV color and scale distribution result per seating point, while Figure 26 shows the summer AFSV color and scale distribution result per seating point. Figure 27 shows the summer average air temperature per measurement position. All these TSV, AFSV, and summer average air temperature data were collected on 4 September 2019. The color of the images shown in Figure 25 defines the hotness and coolness that students felt; the bluer the color, the colder the thermal sensation; the more orange the color, the hotter the thermal sensation. Meanwhile, the color of the images shown in Figure 26 defines the airflow breeziness; the bluer the color, the breezier airflow, and the more orange the color, the more airflow was not felt. Based on the TSV color and scale distribution result per seating point in each classroom shown in Figure 25a–c, there are only a few students who felt hot and mostly felt neutral. The distributions of airflow sensation (Figure 26a–c), which has a dominant bluish color, show that more students felt the airflow than students who did not feel airflow sensation in the classroom. The average air temperature data per class on September 4ᵗʰ, shown in Figure 27, has some blind spots. Some are eliminated because of the extremely high air temperature in the PER zone in the front and middle rows (measurement points ⑦ and ⑥), which is caused by the direct solar radiation effect from the measurement items installation error. The other is point ④ in class B due to the recording data failure.

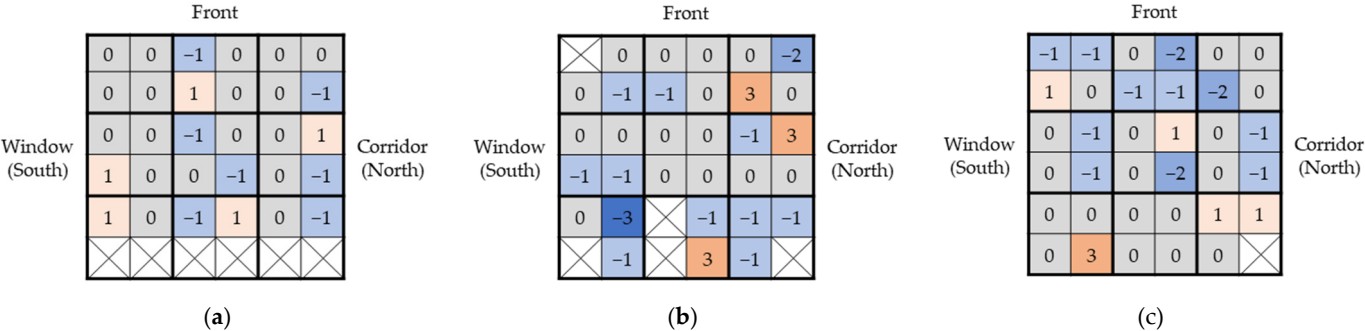

**Figure 25.** Summer TSV result per seating point in (**a**) class A, (**b**) class B, and (**c**) class C.

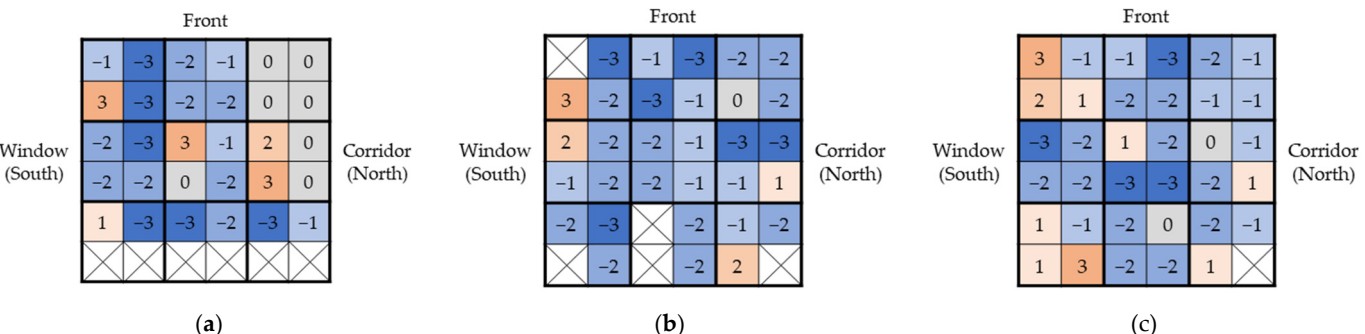

**Figure 26.** Summer AFSV result per seating point in (**a**) class A, (**b**) class B, and (**c**) class C.

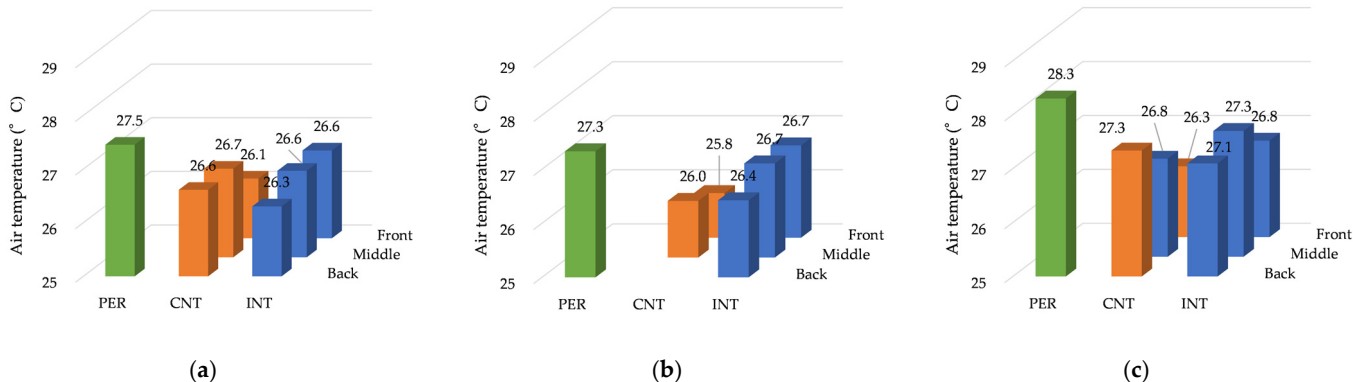

**Figure 27.** Summer average air temperature in (**a**) class A, (**b**) class B, and (**c**) class C.

Table 7 shows the average TSV and AFSV per row and zone. Based on the average per row and zone result of the TSV in class A, the highest TSV was near the window. On the other hand, the lowest ASFV average per zone was in the PER zone, and the second row from the window for the lowest ASFV average per row means that students who sat in that zone felt air flow stronger than in the other zones. Based on the average per row result of TSV in class B, the highest TSV was the third row near the corridor, and the average per zone result of the TSV shows that the INT zone had the highest TSV scale. Figure 27b shows that the air temperature in the INT zone is slightly higher than in the CNT zone. It can be said that the AC airflow blew directly to the CNT zone, which made the air temperature in CNT lower than in the other zones, and the AFSV in the CNT zone was lower than in the other zones.

**Table 7.** TSV and AFSV average per row and zone.

| | | Class A | | | | | | Class B | | | | | | Class C | | | | | | Total Average | | | | | |
|---|---|---|---|---|---|---|---|---|---|---|---|---|---|---|---|---|---|---|---|---|---|---|---|---|---|
| TSV | Row [1] | 0.4 | 0.0 | −0.4 | 0.0 | 0.0 | −0.4 | −0.3 | −1.0 | −0.5 | 0.3 | 0.0 | 0.0 | 0.0 | 0.0 | −0.2 | −0.7 | −0.2 | −0.2 | 0.1 | −0.3 | −0.4 | −0.1 | −0.1 | −0.2 |
| | Zone [2] | 0.2 | | −0.2 | | −0.2 | | −0.6 | | −0.1 | | 0.0 | | 0.0 | | −0.4 | | −0.2 | | −0.1 | | −0.2 | | −0.1 | |
| AFSV | Row [1] | −0.2 | −2.8 | −0.8 | −1.6 | 0.4 | −0.2 | 0.5 | −2.3 | −2.0 | −1.7 | −0.8 | −1.6 | 0.3 | −0.3 | −1.5 | −2.0 | −1.0 | −0.6 | 0.2 | −1.8 | −1.4 | −1.8 | −0.5 | −0.8 |
| | Zone [2] | −1.5 | | −1.2 | | 0.1 | | −0.9 | | −1.8 | | −1.2 | | 0.0 | | −1.8 | | −0.8 | | −0.8 | | −1.6 | | −0.6 | |

[1] Four rows from left to right are window seats to corridor seats. [2] Three zones from left to right are PER, CNT, and INT.

On the other hand, the highest AFSV average per zone scale result was in the PER zone, and the first row from the window for the highest AFSV average per row scale means that students who sat in that zone did not feel air flow as strong as in the other zones. However, similar to class A, the second row from the window had the lowest AFSV average per row. It can be presumed that this position is near the AC equipment that blows air directly to that position. Based on the average per-row result of the TSV in class C, the highest TSV was in the PER zone. Figure 27c shows that the air temperature in the INT zone is slightly higher than in the CNT zone except for the back row. Therefore, it can be said that the AC airflow blew directly to the CNT zone, which made the air temperature in CNT lower than in the other zones, and the AFSV in the CNT zone was lower than in the other zones. On the other hand, the highest AFSV average per zone scale result was in PER zone, and the first row from the window for the highest AFSV average per row scale means that students who sat in that zone did not feel air flow as strong as in the other zones. In terms of the differences from other classes, the third row from the corridor had a minor AFSV average per row.

Figure 28a shows the summer TSV results in each classroom. Students on average answered 0 for TSV, which is "neutral", while students who felt "slightly cool" and "cool" were higher than those who felt slightly warm and warm. On the other hand, the AFSV result shows (Figure 28b) that more students felt "neutral" to "too breezy" than students

who felt "still" to "too still". This indicates that the classrooms had airflow which might have come from the AC or corridor.

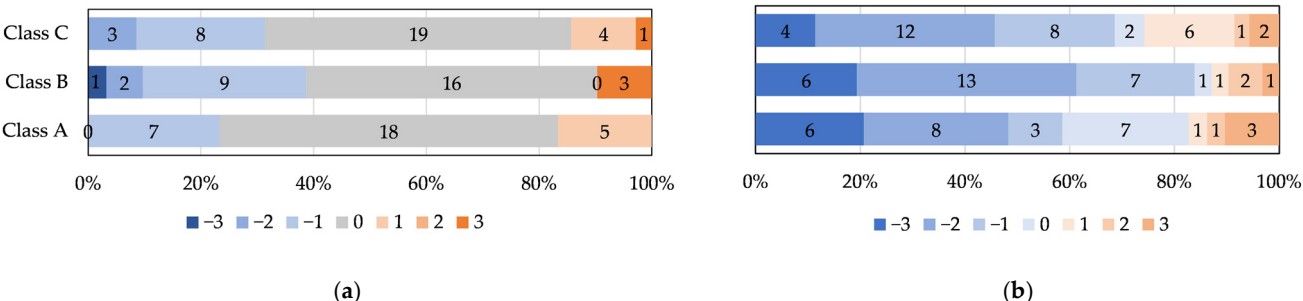

(**a**)  (**b**)

**Figure 28.** Summer questionnaire result per classroom. (**a**) TSV result. (**b**) AFSV result.

Figure 29 shows the correlation between the field measurement result and TSV. The indoor thermal comfort range percentage (based on BESCS) and TSV correlation are shown in Figure 29a. Data on the indoor thermal comfort range percentage based on the BESCS are derived from the average data of the three classrooms on 4 September 2019, when the TSV questionnaire was distributed. The TSV data in this graph are the three classrooms averaged values calculated from the average of four seat positions TSV near each measurement point. It shows that point ⑧ has the smallest TSV value (−0.5) and is correlated to the thermal comfort range percentage based on the BESCS, which has the smallest comfort range percentage (47.6%).

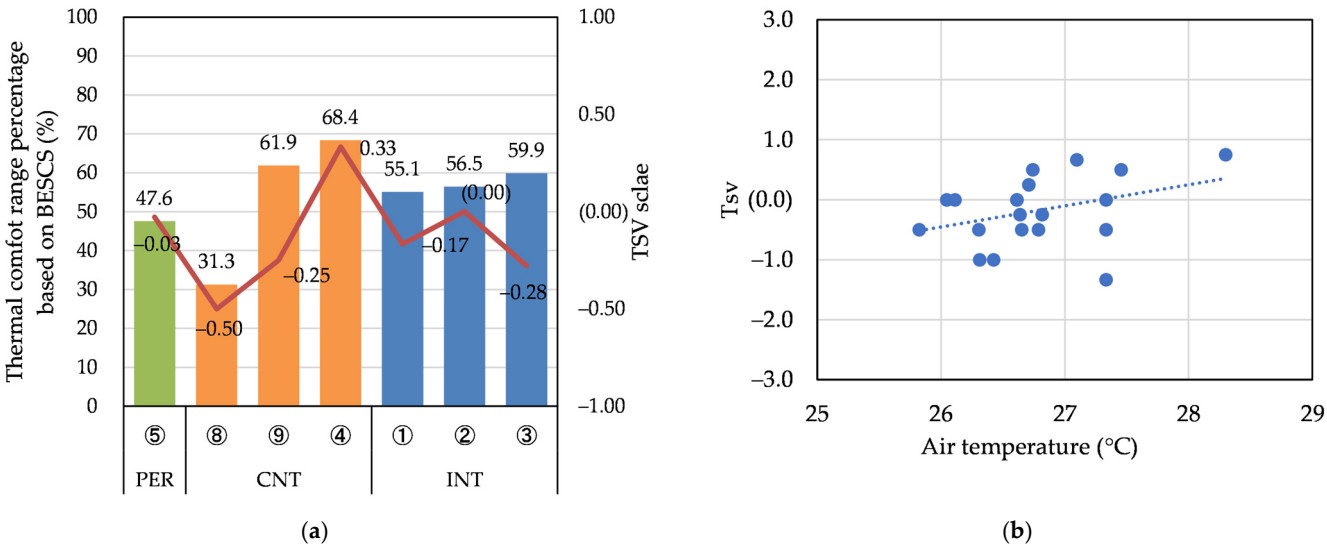

(**a**)  (**b**)

**Figure 29.** The correlation between field measurement result and TSV. (**a**) Indoor thermal comfort range percentage and TSV correlation. (**b**) Summer AT and TSV correlation.

On the other hand, Figure 29b shows there is no significant correlation between the nine points of the one-day average temperature in the three classrooms and the thermal sensation vote of the students present around each measurement point. The simple regression equation was $y = 0.3522x - 99.6118$ and $R^2 = 0.13$. Since the $R^2$ value is 0.13, it can be said that there is a slight correlation between the measured temperature and thermal sensation. This simple regression equation is a positive correlation in which the thermal sensation increases as the measured temperature rises. However, it cannot be a strong argument due to the small value of $R^2$.

## 4. Discussion and Limitations

The AC EU monitoring data results show that the AC EU in the summer was higher than in the winter. Annually, July following September had the highest AC EU in 2019 and 2021, while in 2020, August had the highest AC EU. The AC operating time result in the summer season shows that the longer the AC operates, the higher the AC EU. However, the longest AC operating time annually was in December, following February, with a lower AC EU compared to the summer season. It can be claimed that in addition to the AC operating time, the AC setting temperature influenced the AC EU in the summer. From the monitoring data result, the AC setting temperature in the targeted classrooms was lower than 28 °C, which is the government's recommended value for the AC setting temperature [19]. AC COP in the winter season in Oita City, which is not extremely cold during the winter, also tends to be higher than in the summer, which is caused by this different result in the AC EU in these two seasons. There was high AC EU escalation in 2020–2021 and 2021–2022 because of the COVID-19 pandemic. The school protocol to open windows and doors regularly during a lesson to prevent virus transmission affects the indoor thermal environment, leading to longer AC operating times in the summer and winter and lower AC setting temperature in the summer.

Based on the indoor air temperature analysis result, the highest comfort range percentage of air temperature in each class was at point ③, 84.2% of the total average, and the smallest comfort range percentage of air temperature was at point ⑧, 61.2% of the total average. The total average summer comfort range percentage with the psychometric chart analysis based on the JSEHMS and BESCS also found that point ③ has the highest comfort range (JSEHMS: 8C%; BESCS: 73.8%) and point ⑧ has the smallest comfort range (JSEHMS: 60.5%; BESCS: 47.6%). The total average comfort range percentage between the air temperature analysis and psychometric chart analysis based on the JSEHMS does not have significant differences because the JSEHMS has a looser range of relative humidity ranging from 30 to 80%. However, the psychometric chart analysis based on the BESCS has a different percentage smaller than the JSEHMS. Therefore, it can be concluded that most points have a high comfort range percentage but point ⑧ has a small indoor thermal comfort range based on the BESCS, which is less than 50 % and should be underlined. Point ⑧ also has more air temperature results below 25 °C than above 28 °C, and thus it can be claimed that this point is colder than it should be. The indoor thermal comfort range percentage and TSV correlation result also show that TSV average value at point ⑧ has the smallest TSV value of −0.5. It is correlated to the indoor thermal comfort range percentage based on the BESCS of point ⑧ on the day when the TSV questionnaire was conducted, with the smallest percentage of 31.3%. Point ⑧, which has the smallest range of indoor thermal comfort percentage, has a higher percentage of air temperature below 25 °C than the other points. From the TSV value and air temperature result, it can be claimed that the AC makes the room colder than it should be at this measurement point.

Deepening knowledge about zoning is a promising action to achieve many HVAC system design goals that can positively impact the labor of the designers of these types of systems [42]. This research finds that the PER, CNT, and INT zone correlations show an air temperature difference pattern in each zone. The PER zone has the highest air temperature, followed by the INT zone, and the CNT zone has the lowest air temperature. This also confirmed the TSV and AFSV results, which found that the average TSV in the CNT zone had the lowest value. The CNT zone low air temperature is caused by the airflow of AC, which is strengthened by the result of the average AFSV value in CNT being the lowest compared to the other zones (Table 5). Point ⑤, which is the measurement point in the PER zone, had the highest percentage of air temperature above 28 °C than other points. It can be claimed that the PER zone in the summer had a higher temperature than the other zones. To decrease direct solar radiation in the PER zone area, a typical classroom in almost every junior high school has curtains as solar shielding. However, direct solar radiation still penetrates, and closing the curtains does not necessarily eliminate the effects of solar radiation.

As mentioned in the air temperature result, the PMV is not measured or calculated during the measurement time due to the complexity of measurement items installation. However, this research has conducted a questionnaire distribution to assess the students' thermal sensation, which has the range of the sensation that complied to a PMV value from −3 (cold) to +3 (hot), as shown in Table 6. Based on the summer TSV result, although students mostly felt "neutral", students who felt "slightly cool" and "cool" were higher than those who felt "slightly warm" and "warm". Therefore, it can be claimed that an AC setting temperature lower than 28 °C in the summer can affect the subjective thermal sensation felt by students. Therefore, it is suggested that the government recommended AC setting temperature for classrooms in summer, at 28 °C [19], should be kept as close as possible and not much lower than 28 °C to promote AC energy-saving in the summer. In addition, the indoor air temperature had a high difference from the outdoor air temperature (Figure 23a), which can lead the heat shock, which is generated from zonal temperature differences [43]. However, the research on the maximum value standard of the air temperature difference between outdoor and indoor, which is acceptable for young students' health, has not been progressing. This could be an important future issue to be further investigated.

Although this research did not measure air velocity as one of the thermal comfort parameters, the AFSV questionnaire was distributed to assess the air velocity subjectively from students' senses. The AFSV result shows that more students felt "neutral" to "too breezy" than students who felt "still" to "too still". In addition, as defined in Table 4, the air velocity assumption value to determine the PMV value and air temperature comfort range, 0.2 m/s, might be considered to correspond to this AFSV questionnaire result.

The limitation of this research in indoor air temperature and relative humidity is the measurement errors that occurred in some measurement items. The measurement items at points ⑥ and ⑦ in each classroom are excluded due to directed exposure to solar radiation. The measurement item at point ④ in class B had measurement failure from the beginning to the end of the measurement times. The measurement item at point ⑤ in class B had measurement failure from September 5th to 9th. The last is the measurement item at point ⑩ in class C, which failed to measure on September 9th from the beginning to 12:10. The other limitation is the accuracy of TR-72NW, as the measurement item, ±0.5 °C ±5%RH (at 25 °C, 50%RH). The energy consumption in the AC unit limitation is the uncertainty of the AC units' capacities and COP.

## 5. Conclusions

Based on the sensitive analysis study, we can conclude the following:

(1) It found that the AC EU in the summer was higher than in the winter, even though the AC operation time in the winter was slightly higher than in the summer. In addition, it was found that the AC EU excessively increased from 2020 to 2022 compared to 2019 because of the impact of the COVID-19 pandemic.

(2) It found that in addition to the AC operating times, the AC setting temperature had a great impact on the AC EU.

(3) Based on the comfort range percentage and questionnaire result, each classroom achieved indoor thermal comfort.

(4) It found that point ⑧ had the smallest indoor thermal comfort percentage and had more percentage of air temperatures below 25 °C, which is colder than the comfort range temperature.

(5) The CNT zone had a slightly colder temperature than the other zones, which was caused by airflow from the AC.

(6) It found that most students felt "neutral", and the total number of students who felt "slightly cool" and "cool" was more than the students who felt "slightly warm" and "warm".

(7) Similar to the measurement result, the total average TSV result found that students who sat near point ⑧ felt colder than at other points. It can be claimed that the AC directly affected thermal comfort to this point.

(8)　It found that the PER zone had the highest percentage of air temperature above 28 °C, and thus it can be claimed that thermal comfort in the PER zone was hotter than in the other zones. This is confirmed by the result of TSV in classes A and C, which had the highest TSV value.

(9)　It also found that students who sat in the CNT zone felt colder and more breezy air than in the other zones.

These results further contribute to the future of the profound thinking of the energy-saving strategy, such as the AC setting temperature, as one of the major impacts of AC energy-saving. Based on the measurement result and questionnaire result, the classrooms generally have reached a comfort thermal range in each classroom in the summer, except at point ⑧, due to the low AC setting temperature (below 28 °C). To optimize the energy-saving strategy, it is suggested to maintain the AC setting temperature recommendation by the government in the summer (28 °C). However, it will be a major consideration for indoor comfort if the AC is set to 28 °C in the summer. The challenge of finding the midpoint of the indoor thermal comfort with a lower AC EU, especially in summer must still be considered. These findings also show that the seating layout, AC layout, and AC setting temperature must be considered to achieve indoor thermal comfort and promote AC energy-saving.

**Author Contributions:** Conceptualization, D.S. and Y.R.; data curation, D.S. and K.E.; formal analysis, D.S.; investigation, K.E.; methodology, D.S.; project administration, Y.R.; resources, K.E.; supervision, Y.R. and W.G.; validation, D.S., Y.R. and D.N.; visualization, D.S.; writing—original draft, D.S.; writing—review and editing, D.S. and D.N. All authors have read and agreed to the published version of the manuscript.

**Funding:** This research received no external funding.

**Data Availability Statement:** The data presented in this study are available on request from the corresponding author. The data are not publicly available due to privacy matter.

**Acknowledgments:** The authors wish to acknowledge the help of the PFI monitoring data used in this research provided by the Oita City municipal office.

**Conflicts of Interest:** The authors declare no conflict of interest.

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
