# Peer review of "Energy-Use and Indoor Thermal Performance in Junior High School Building after Air-Conditioning Installation with the Private Finance Initiative"

_buildings, doi:10.3390/buildings13020455_

Round 1

Reviewer 1 Report

The reviewed manuscript is at an average level when it comes to the originality of the presented solutions and formulated final conclusions, but it may meet with the interest of readers and its results can be used in the design and operation of air conditioning systems in school rooms. The manuscript presents the results of experimental studies and student thermal feedback (TSV – Thermal Sensation Vote) surveys, as well as data obtained from monitoring the performance of air conditioning systems in schools implemented as part of the "Private Finance Initiative" in recent years in Japan.

Despite the fact that the manuscript has only a descriptive form and there are no mathematical models simulating temperature distributions and air flow in air-conditioned school rooms, it brings to the discussion a number of comments on the factors affecting the energy efficiency of air conditioning systems in school rooms. The conclusions of the manuscript may influence the more optimal design and operation of air-conditioning systems in school rooms in terms of reducing the energy consumption of these systems.

The manuscript presented in its present form requires some changes to be eligible for publication, in particular as regards linguistic and stylistic correctness. I also strongly recommend the introduction of a list of abbreviations used in the text, which will significantly improve the readability of the manuscript.

Author Response

Response to Reviewer 1 Comments

The reviewed manuscript is at an average level when it comes to the originality of the presented solutions and formulated final conclusions, but it may meet with the interest of readers and its results can be used in the design and operation of air conditioning systems in school rooms. The manuscript presents the results of experimental studies and student thermal feedback (TSV – Thermal Sensation Vote) surveys, as well as data obtained from monitoring the performance of air conditioning systems in schools implemented as part of the "Private Finance Initiative" in recent years in Japan.

Response: Thank you for the time and effort dedicated to providing feedback and suggestion on our manuscript. We really hope that our manuscript may meet the interest of the readers of the Buildings journal.

Despite the fact that the manuscript has only a descriptive form and there are no mathematical models simulating temperature distributions and air flow in air-conditioned school rooms, it brings to the discussion a number of comments on the factors affecting the energy efficiency of air conditioning systems in school rooms. The conclusions of the manuscript may influence the more optimal design and operation of air-conditioning systems in school rooms in terms of reducing the energy consumption of these systems.

Response: Thank you for the insightful comment. This research focuses on analyzing the PFI data monitoring, field measurement and questionnaire results. We are aware of conducting a simulation of the airflow and air temperature distribution based on the actual field measurement data result in future research. 

The manuscript presented in its present form requires some changes to be eligible for publication, in particular as regards linguistic and stylistic correctness. I also strongly recommend the introduction of a list of abbreviations used in the text, which will significantly improve the readability of the manuscript.

Response: Thank you for the suggestion. Incorporating the suggestions, we modified our manuscript for improvements, such as linguistic and stylistic correctness, measurement data result addition to support the result, the improvement of the discussion part, the limitation, and the result of the research. All modifications in the manuscript have been using “track changes” in Microsoft Word. The measurement data results that have been added are as follows:

  1. Figure 7. Outdoor summer air temperature and relative humidity in measurement time. (page 8)
  2. Figure 23. AT difference (a) Air temperature difference between outdoor and indoor Air temperature. (page 17)
  3. Figure 24. Average air temperature distribution per level measurement (10 cm, 70 cm, and 280 cm) in each classroom. (page 18)

We also added the abbreviation list and its meaning in Table 1 on page 3. 

Reviewer 2 Report

This article studies the thermal comfort inside high school classrooms in the City of Oita, Japan, by means of measurements of indoor air temperatures and relative humidity and assessing the students' comfort with questionnaires. Overall, the subject of research is interesting as it addresses several important issues: prevention of heat strokes due to summer overheating, but also excessive cooling of the indoor ambient. Improvements are necessary primarily in the section concerning results analysis and discussion, by addressing the comments below:

1. Line 77: the figure reference should be for Figure 2a.

2. The thermal comfort analysis includes measurements of indoor air temperatures and relative humidity, and a questionnaire to determine the students' thermal and air flow sensations. Did you consider also using the standard method for the predicted mean vote (PMV) and predicted percentage of dissatisfied (PPD), described in ANSI/ASHRAE Standard 55?

3. In the PMV/PPD method, beside indoor air temperature and humidity, other parameters are also evaluated such as the occupants' metabolic rate and clothing insulation, mean radiant temperature and air velocity. Your method is based primarily on measurements of air temperature and humidity. The neglected thermal comfort parameters would have changed the results and conclusions if they were included in the analysis? You did not measure the real air flow velocity, however, the subjective air flow sensation was assessed by questionnaires distributed to occupants? Please add a discussion in the article.

4. The questionnaire was distributed on September 4, 2019 while measurements were collected between August 27 and September 9, 2019. What was the outdoor weather and indoor conditions on the day of the questionnaire? These conditions were representative for the entire summer period?

5. What is the EHP = 9.97 MJ/kWh in table 2? How was this value obtained?

6. Figure 2a shows that each class has two AC units under the ceiling. What is the cooling and heating capacities (in kW) of these units, what is the coefficient of performance (COP)? Line 149 says that the AC units use electricity and LPG as heat sources. However, the heat source in winter period should be outdoor air, while electricity and LPG should be used for running the compressor?

7. How was the AC energy use (EU) determined? What is the data source (energy meters, billed energy consumption, electricity and LPG consumption)? If the data was obtained from energy bills, then how was the AC energy use distinguished from energy use for lighting, appliances, cooking?

8. Section 3.3.1. mentions the "hunting phenomenon" for the large oscillations of indoor air temperatures (figures 7-9). However, this term is not used in the literature.

9. Line 304 says that the data for point ⑤ in class C is neglected due to measurement errors. However, figures 14 and 15 are missing only data point ④ for class B while data point ⑤ for class C is supplied.

10. The results show that the air temperature in the perimeter zone (PER) is higher than in the other zones of the classrooms. This is due to the vicinity of the PER zone to windows which allow for solar heat gains. The classrooms did not have any protection from sunlight in summer (solar blinds or shutters)?

11. Figures 19-21: the suction temperature is measured at a height of 280 cm (line 115) while the indoor temperature is measured at 70 cm or 100 cm (line 141). The suction temperature is generally higher than the indoor temperature, which suggests that the classrooms exhibit vertical temperature stratification is summer with higher air temperatures under the ceiling and lower air temperature right above the floor. Figure 22 shows the temperature difference between the height of 280 cm and 100 cm? How does this temperature stratification affect the thermal comfort in the classrooms?

12. Generally, the results show that the classrooms are excessively cooled in the summer period as a consequence of the too low cooling set-point (below 28 °C). This trend is also seen in residential and public/office buildings. What would be your suggestion to occupant and local authorities to prevent thermal discomfort and excessive energy consumption due to indoor over-cooling in summer?

13. In the Introduction, you discuss about the heat stroke problem due to excessive summer temperatures. As a remedy, AC units were installed in high schools in Japan (lines 36-38). On the other side, your results show that high school classrooms can have excessive cooling and too low temperatures in summer, which again could lead to health issues related to large temperature difference between indoor and outdoor spaces?

14. What is the measurement uncertainty for indoor air temperature and relative humidity, and energy consumption in AC units?

Author Response

Response to Reviewer 2 Comments

This article studies the thermal comfort inside high school classrooms in the City of Oita, Japan, by means of measurements of indoor air temperatures and relative humidity and assessing the students' comfort with questionnaires. Overall, the subject of research is interesting as it addresses several important issues: prevention of heat strokes due to summer overheating, but also excessive cooling of the indoor ambient.

Response: We agreed with the reviewer’s assessment to improve the manuscript, and we tried to address most of the issues mentioned by the reviewers and editor.

  1. Line 77: the figure reference should be for Figure 2a.

Response: Thank you for pointing out this mistake. The figure reference has been corrected in line 78.

  1. The thermal comfort analysis includes measurements of indoor air temperatures and relative humidity, and a questionnaire to determine the students' thermal and air flow sensations. Did you consider also using the standard method for the predicted mean vote (PMV) and predicted percentage of dissatisfied (PPD), described in ANSI/ASHRAE Standard 55?

Response: Thank you for pointing this out. We think this is an excellent suggestion. In this study, "Predicted mean vote (PMV)," as a standard method to measure indoor thermal comfort described by ASHRAE Standard 55, could not be calculated, which is the limitation of this research. This experiment did not measure other parameters, such as PMV or globe temperature and air velocity, due to complex sensor installation and re-straining teaching and learning activities intervention measuring other parameters. Otherwise, the air temperature and relative humidity were measured by small-size measurement items, which did not interfere with students' activities. However, in this study, as mentioned, the air temperature range, 25°C to 28°C, is considered the comfort range and is based on the revision of the school's environmental hygiene standards. Besides, it was also obtained from acceptable Predicted Mean Vote (PMV) by "International Organization for Standardization (ISO)" 7730:2005 as ranging for existing buildings between -0.7 and +0.7. Calculation data assumed to determine range standard with acceptable PMV by ISO are shown in Table 4. This explanation has been added to the revised manuscript lines 281-293. In addition, although PMV is not measured or calculated, this research has conducted a questionnaire distribution to assess the students' thermal sensation, which has the range of the sensation that complied to PMV value from -3 (cold) to +3 (hot), as shown in Table 6. This explanation has been added in discussion part lines 581-585.

  1. In the PMV/PPD method, beside indoor air temperature and humidity, other parameters are also evaluated such as the occupants' metabolic rate and clothing insulation, mean radiant temperature and air velocity. Your method is based primarily on measurements of air temperature and humidity. The neglected thermal comfort parameters would have changed the results and conclusions if they were included in the analysis? You did not measure the real air flow velocity; however, the subjective air flow sensation was assessed by questionnaires distributed to occupants? Please add a discussion in the article.

Response: Thank you for pointing this out. Although this research did not measure air velocity as one of the thermal comfort parameters, the AFSV questionnaire was distributed to assess the air velocity subjectively from students’ senses. The AFSV result shows that students who felt “neutral” to “too breezy” are more than students who felt “still” to “too still”. In addition, as defined in Table 4, the air velocity assumption value to determine PMV value and air temperature comfort range, 0.2 m/s, might be considered to correspond to this AFSV questionnaire result. This explanation has been added in discussion part line 597-603.

  1. The questionnaire was distributed on September 4, 2019 while measurements were collected between August 27 and September 9, 2019. What was the outdoor weather and indoor conditions on the day of the questionnaire? These conditions were representative for the entire summer period?

Response: Thank you for pointing this out. TSV and “air flow sensation vote (AFSV)” questionnaires were distributed on September 4th, 2019. In this research, September could represent the entire summer period because the rainy season lasts from the beginning of June to mid-July, while August is the summer holiday. Besides, when the questionnaire was conducted, the students were informed to fill out the questionnaire as they generally felt about their indoor thermal sensation in the last seven days prior to September 4th, 2019. In addition, the outdoor air temperature on this day reached above 30°C after 11:00 (Figure 7), which fitted to the standard of summer climate. Therefore, it could represent a sunny day at the peak of summer. This explanation has been added in lines 445-453. In addition, Figure 7, as an outdoor air temperature and relative humidity per hour from September 4th to 9th has been added on page 8.

  1. What is the EHP = 9.97 MJ/kWh in table 2? How was this value obtained?

Response: Thank you for pointing this out. EHP= 9.97MJ/kWh is unit calorific value. It is obtained from The Enforcement Regulations of the Law Concerning the Rational Use of Energy, which stipulate the numerical value for converting electric power into energy consumption. Unit calorific value for daytime electricity is 9.97 MJ/kWh. This explanation has been added on page 5, lines 160 to 164.

  1. Figure 2a shows that each class has two AC units under the ceiling. What is the cooling and heating capacities (in kW) of these units, what is the coefficient of performance (COP)? Line 149 says that the AC units use electricity and LPG as heat sources. However, the heat source in winter period should be outdoor air, while electricity and LPG should be used for running the compressor?

Response: Thank you for pointing this out. Unfortunately, we do not obtain data about the indoor units' exact cooling and heating capacities and coefficient of performance. Thank you for pointing out the mistake of LPG. LPG is used for running the compressor, not as a heat source. We have corrected the statement in line 155.

  1. How was the AC energy use (EU) determined? What is the data source (energy meters, billed energy consumption, electricity, and LPG consumption)? If the data was obtained from energy bills, then how was the AC energy use distinguished from energy use for lighting, appliances, cooking?

Response: Thank you for pointing this out. We are aware that the detail of PFI method and data obtaining information were not clear enough. The PFI method had only managed the energy of AC equipment for heating and cooling, while other energy consumption management had already been conducted before the PFI method was introduced in the schools. The PFI method had only managed the energy of AC equipment, while other energy consumption management had already been conducted before the PFI method was introduced in the schools. This explanation has been added in lines 62-65. AC EU, in power consumption (kWh) and gas (LPG) consumption (m3), was determined by the PFI method that collected only AC EU, not other`s energy use. AC EU is calculated from primary data obtained in kWh (electric) and m3 (gas) and converted to GJ with each heat source conversion (unit calorific value), which is shown in Table 3. This explanation has been added in lines 156-160.

  1. Section 3.3.1. mentions the "hunting phenomenon" for the large oscillations of indoor air temperatures (figures 7-9). However, this term is not used in the literature.

Author response: Thank you for pointing this out and for the suggestion to use literature reference. The hunting phenomena is known also as an oscillatory behavior. This oscillatory behavior is determined as the common HVAC system disturbances, which control the AC setting temperature and suction air temperature about 1.5°C-2°C. It may happen because of the outdoor temperature changes, or the occupancy of the rooms being conditioned. This explanation has been added in lines 254-257.

  1. Line 304 says that the data for point ⑤ in class C is neglected due to measurement errors. However, figures 14 and 15 are missing only data point ④ for class B while data point ⑤ for class C is supplied.

Response: Thank you for pointing this out. Point ⑤ as the point in the perimeter zone was above 70% comfort based on JSEHMS (Figures 15b, 15c) and above 65% comfort based on BESCS (Figures 16b, 16c) except for Class C (Figures 15d, 16d). Data for point ⑤ in class C supplied in the graph is only the data on September 4th. However, it won’t be further analyzed due to the lack of data on other measurement days. Meanwhile, data point â‘£ for class B is not supplied because there is no data obtained on all measurement days due to measurement error. We changed this explanation in lines 345-349.  

  1. The results show that the air temperature in the perimeter zone (PER) is higher than in the other zones of the classrooms. This is due to the vicinity of the PER zone to windows which allow for solar heat gains. The classrooms did not have any protection from sunlight in summer (solar blinds or shutters)?

Response: Thank you for pointing this out. To decrease direct solar radiation in the PER zone area, a typical classroom in almost every junior high school has curtains as solar shielding. However, direct solar radiation still penetrates, and closing the curtains does not necessarily eliminate the effects of solar radiation. This explanation has been added in lines 577-580.

  1. Figures 19-21: the suction temperature is measured at a height of 280 cm (line 115) while the indoor temperature is measured at 70 cm or 100 cm (line 141). The suction temperature is generally higher than the indoor temperature, which suggests that the classrooms exhibit vertical temperature stratification is summer with higher air temperatures under the ceiling and lower air temperature right above the floor. Figure 22 shows the temperature difference between the height of 280 cm and 100 cm? How does this temperature stratification affect the thermal comfort in the classrooms?

Response: Thank you for pointing this out. Figure 22, now revised to be Figure 23 (b), shows that the air temperature difference between the air temperature actual field measurement data and air temperature data collected with PFI is about 0.2°C to 3.1°C. Air temperature actual field measurement data is the air temperature average data of points â‘ , â‘¡, â‘¢, â‘£, ⑤, â‘§, and ⑨, with 70 cm and 100 cm height of the measurement items, while air temperature data collected with the PFI method is the suction temperature in AC level height (2.8 m). On the other hand, Figure 24 shows the air temperature distribution by the measurement level, 10 cm, 70 cm, and 280 cm. The 10 cm measurement level is obtained from point â‘© data, and 70 cm from point ⑨ data, which is located in the middle of the room and might be considered representative of average room air temperature. The air temperature data shown in the graph are averaged data from September 4th to 9th at 08:00-16:00. However, data in level 10 cm in class C is the average air temperature data from September 9th, 12:20, to the end of the measurement. It arose due to the measurement item in point â‘© in class C had failed to measure on September 9th from the be-ginning to 12:10. Based on the air temperature distribution result, the difference between air temperature at 70 cm level and 10 cm level did not exceed more than 1°C. Therefore, it can be stated that there is no extreme temperature stratification. However, the air temperature difference between the 280 cm level and 70 cm average exceeds more than 1°C. It is not only caused by the high position but also caused by the position of the AC above the southern windows, which are the warmest side in the room, caused by solar radiation effects. The air temperature difference between the suction air temperature at 280 cm and room temperature at 70 cm level does not necessarily affect the thermal comfort of the high position of the suction measurement level, which is not the level of the learning activities. However, this AC suction air temperature is also used for indoor air temperature data monitoring. Therefore, if indoor thermal monitoring in schools with the PFI method without actual measurement is conducted, this temperature difference between PFI air temperature monitoring data and actual field measurement results must be considered This explanation is on page 18, lines 414-440. Figure 24, as an average air temperature distribution per level measurement (10 cm, 70 cm, and 280 cm) in each classroom has been added on page 18.

  1. Generally, the results show that the classrooms are excessively cooled in the summer period as a consequence of the too low cooling set-point (below 28 °C). This trend is also seen in residential and public/office buildings. What would be your suggestion to occupant and local authorities to prevent thermal discomfort and excessive energy consumption due to indoor over-cooling in summer?

Response: Thank you for pointing this out. Based on comfort range percentage and questionnaire result, each classroom achieved indoor thermal comfort. This explanation has been added to the conclusion part number (3) line 622. In addition, based on the measurement result and questionnaire result, the classrooms generally have reached comfort thermal range in each classroom in summer, except in point â‘§, due to the low AC setting temperature (below 28°C). To optimize energy-saving strategy, it is suggested to maintain the AC setting temperature recommendation from the government in summer (28°C). However, it will be a major consideration for indoor comfort if the AC set to 28°C in summer. This explanation has been added in lines 643-648. Excessive may not be the most proper word to explain the result of this paper, because the only point which has excessive cold temperature is point â‘§, while other points have more than 60% comfort percentage from total measurement data.

  1. In the Introduction, you discuss about the heat stroke problem due to excessive summer temperatures. As a remedy, AC units were installed in high schools in Japan (lines 36-38). On the other side, your results show that high school classrooms can have excessive cooling and too low temperatures in summer, which again could lead to health issues related to large temperature difference between indoor and outdoor spaces?

Response: Thank you for pointing this out. Figure 23(a) shows the outdoor and indoor air temperature differences in each classroom. The difference is from -1.2°C to 7.5°C, with a higher difference mostly hit after 11:00 to 15:00. This explanation has been added in lines 407-409. The indoor air temperature had a high difference from outdoor air temperature, which can lead the heat shock, which is generated from zonal temperature differences. However, the research about how much the maximum value standard of the air temperature difference between outdoor and indoor, which is acceptable for young students’ health, has not been progressing. This could be an important future issue to be further investigated. This explanation has been added in lines 591-592. The outdoor and indoor air temperature graph is also added on page 17.

  1. What is the measurement uncertainty for indoor air temperature and relative humidity, and energy consumption in AC units?

Response: Thank you for pointing this out. The limitation of this research in indoor air temperature and relative humidity is the measurement errors that occurred in some measurement items. Measurement items in points â‘¥ and ⑦ in each classroom are neglected due to directed exposure to solar radiation. The measurement item in point â‘£ in class B had measurement failure from the beginning to the end of the measurement times. Measurement item in point ⑤ class B had measurement failure from September 5th to 9th. The last is the measurement item in point â‘© in class C had failed to measure on September 9th from the beginning to 12:10. The other limitation is the accuracy of TR-72NW, as measurement item, ±0.5°C ±5%RH (at 25°C, 50%RH). The energy consumption in AC unit limitation is the uncertainty of the AC units’ capacities and COP. This explanation has been added in lines 604-613.

Round 2

Reviewer 2 Report

The authors have supplied detailed explanations to all the comments and suggestions. The necessary improvements have been made into the manuscript.